# Calibrated and Conformal Propensity Scores for Causal Effect Estimation

**Shachi Deshpande**[1]                    **Volodymyr Kuleshov**[1]

[1]Dept of Computer Science, Cornell University and Cornell Tech, New York, NY, USA

## Abstract

Propensity scores are commonly used to estimate treatment effects from observational data. We argue that the probabilistic output of a learned propensity score model should be calibrated—i.e., a predictive treatment probability of 90% should correspond to 90% individuals being assigned the treatment group—and we propose simple recalibration techniques to ensure this property. We prove that calibration is a necessary condition for unbiased treatment effect estimation when using popular inverse propensity weighted and doubly robust estimators. We derive error bounds on causal effect estimates that directly relate to the quality of uncertainties provided by the probabilistic propensity score model and show that calibration strictly improves this error bound while also avoiding extreme propensity weights. We demonstrate improved causal effect estimation with calibrated propensity scores in several tasks including high-dimensional image covariates and genome-wide association studies (GWASs). Calibrated propensity scores improve the speed of GWAS analysis by more than two-fold by enabling the use of simpler models that are faster to train.

## 1  INTRODUCTION

This paper studies the problem of inferring the causal effect of an intervention from observational data. For example, consider the problem of estimating the effect of a treatment on a medical outcome or the effect of a genetic mutation on a phenotype. A key challenge in this setting is confounding—e.g., if a treatment is only given to sick patients, it may paradoxically appear to trigger worse outcomes [Greenland et al., 1999, VanderWeele, 2006].

Propensity score methods are a popular tool for correcting for confounding in observational data [Rosenbaum and Rubin, 1983, D'Agostino, 1998, Imbens, 2000, Lanza et al., 2013]. However, propensity score methods can become unreliable when their predictive model outputs incorrect treatment assignment probabilities [Kang and Schafer, 2007, A. Smith and E. Todd, 2005, Lenis et al., 2018]. For example, when the propensity score model is overconfident (a known problem with neural network estimators Guo et al. [2017]), predicted assignment probabilities can be too small [Tan, 2017], which yields a blow-up in the estimated causal effects. More generally, propensity score weighting stands to benefit from accurate uncertainty estimation Kallus [2020].

This work argues that propensity score methods can be improved by leveraging calibrated uncertainty estimation in treatment assignment models. Intuitively, when a calibrated model outputs a treatment probability of 90%, then 90% of individuals with that prediction should be assigned to the treatment group [Platt, 1999, Kuleshov et al., 2018]. We argue that calibration is a necessary condition for propensity score models and it also addresses the aforementioned problems of model overconfidence.

Off-the-shelf propensity score models are typically uncalibrated Kallus [2020]; our work introduces algorithms that provably enforce uncertainty calibration in these models. Post-processing of propensity weights is often done via trimming [Crump et al., 2009, Fan Li and Zaslavsky, 2018], but this can introduce bias by eliminating information from propensity weights below a pre-selected trimming threshold [Li et al., 2018]. Propensity score calibration reduces variance from extreme propensity weights without introducing bias from trimming thresholds. Approaches that balance covariates during optimization to obtain propensity weights have demonstrated theoretical and empirical advantages in causal effect estimation [Hainmueller, 2012, Chan et al., 2016, Zhao, 2017, Zubizarreta, 2015, Ning et al., 2018], but choosing appropriate covariate balancing conditions requires substantial knowledge of the observational study [Ben-Michael et al., 2021]. Uncertainty calibration is simpler to implement and can be combined

with any base propensity model and methods like trimming or covariate balancing without changing the model optimization procedure.

In summary, this paper makes the following contributions: (1) we provide formal arguments that establish calibration as a necessary condition for unbiased treatment effect estimation, prove the reduction of variance by avoiding extreme propensity weights and show improved error bounds on the causal effect estimates by enforcing calibration; (2) we propose simple algorithms that enforce calibration; (3) we provide theoretical guarantees on the calibration and regret of these algorithms; (4) we demonstrate the effectiveness of calibrated propensities in several tasks and show improvement in the speed of high-dimensional genome-wide association studies (GWASs) by more than two-fold.

## 2 BACKGROUND

**Notation** Formally, we are given an observational dataset $\mathcal{D} = \{(x^{(i)}, y^{(i)}, t^{(i)})\}_{i=1}^n$ consisting of $n$ units, each characterized by features $x^{(i)} \in \mathcal{X} \subseteq \mathbb{R}^d$, a binary treatment $t^{(i)} \in \{0, 1\}$, and a scalar outcome $y^{(i)} \in \mathcal{Y} \subseteq \mathbb{R}$. We assume $\mathcal{D}$ consists of i.i.d. realizations of random variables $X, Y, T \sim P$ from a data distribution $P$. Although we assume binary treatments and scalar outcomes, our approach naturally extends beyond this setting. The feature space $\mathcal{X}$ can be any continuous or discrete set.

### 2.1 CAUSAL EFFECT ESTIMATION USING PROPENSITY SCORING

We seek to estimate the true effect of $T = t$ in terms of its average treatment effect (ATE).

$$Y[x, t] = \mathbb{E}[Y | X = x, \mathrm{do}(T = t)] \quad (1)$$
$$\mathrm{ATE} = \mathbb{E}[Y[x, 1] - Y[x, 0]], \quad (2)$$

where $\mathrm{do}(\cdot)$ denotes an intervention [Pearl et al., 2000]. We assume strong ignorability, i.e., $(Y(0), Y(1)) \perp T | X$ and $0 < P(T|X) < 1$, for all $X \in \mathcal{X}, T \in \{0, 1\}$, where $Y(0)$ and $Y(1)$ denote potential outcomes. We also make the stable unit treatment value assumption (SUTVA), which states that there is a unique value of outcome $Y_i(t)$ corresponding to unit $i$ with input $x_i$ and treatment $t$ [Rosenbaum and Rubin, 1983]. Under these assumptions, the propensity score defined as $e(X) = P(T = 1|X)$ satisfies the conditional independence $(Y(0), Y(1)) \perp T|e(X)$ [Rosenbaum and Rubin, 1983]. Propensity score also acts as a balancing score, i.e. $X \perp T|e(X)$. Thus, ATE can be expressed as $\tau = \mathbb{E}\left(\frac{TY}{e(X)} - \frac{(1-T)Y}{1-e(X)}\right)$. The Inverse Propensity of Treatment Weight (IPTW) estimator uses an approximate model $Q(T = 1|X)$ of $P(T = 1|X)$

to produce an estimate $\hat{\tau}$ of the ATE, which is computed as

$$\hat{\tau}_1 = \frac{1}{n} \sum_{i=1}^n \left( \frac{t^{(i)} y^{(i)}}{Q(T=1|x^{(i)})} - \frac{(1-t^{(i)}) y^{(i)}}{1 - Q(T=1|x^{(i)})} \right).$$

The Augmented Inverse Propensity Score Weight (AIPW) estimator uses an outcome model $f(X, T)$ to approximate the potential outcome $Y[X, T]$, thus computing ATE as $\hat{\tau}_2 = \hat{\tau}_1 + \frac{1}{n} \sum_{i=1}^n \left[ f(x_i, T = 1)\left(1 - \frac{t_i}{Q(T=1|x^{(i)})}\right) - f(x_i, T = 0)\left(1 - \frac{(1-t_i)}{1-Q(T=1|x^{(i)})}\right) \right]$. This doubly robust estimator can produce accurate ATE estimates when either the propensity model or the outcome model is correctly specified [Robins et al., 1994].

### 2.2 CALIBRATED AND CONFORMAL PREDICTION

This paper seeks to evaluate and improve the uncertainty of propensity scores. A standard tool for evaluating predictive uncertainties is a proper loss (or proper scoring rule) $L : \Delta_{\mathcal{Y}} \times \mathcal{Y} \to \mathbb{R}$, defined over the set of distributions $\Delta_{\mathcal{Y}}$ over $\mathcal{Y}$ and a realized outcome $y \in \mathcal{Y}$. Examples of proper losses include the L2 or the log-loss. It can be shown that a proper score is a sum of the following terms [Gneiting et al., 2007]: proper loss = calibration − sharpness + irreducible term.

**Calibration.** Intuitively, calibration means that a 90% confidence interval contains the outcome about 90% of the time. Sharpness means that confidence intervals should be tight. Maximally tight and calibrated confidence intervals are Bayes optimal. In the context of propensity scoring methods for binary treatments, we say that a propensity score model $Q$ is calibrated if the true probability of $T = 1$ conditioned on predicting a probability $p$ matches the predicted probability:

$$P(T = 1 \mid Q(T = 1|X) = p) = p \ \ \forall p \in [0, 1] \quad (3)$$

**Calibrated and Conformal Prediction.** Out of the box, most models $Q$ are not calibrated. Calibrated and conformal prediction yield calibrated forecasts by comparing observed and predicted frequencies on a hold-out dataset [Shafer and Vovk, 2007, Kuleshov et al., 2018, Angelopoulos and Bates, 2021, Vovk et al., 2005].

## 3 CALIBRATED PROPENSITY SCORES

We start with the observation that a good propensity scoring model $Q(T|X)$ must not only correctly output the treatment assignment, but also accurately estimate predictive uncertainty. Specifically, the *probability* of the treatment assignment must be correct, not just the class assignment. While a Bayes optimal $Q$ will perfectly estimate uncertainty, suboptimal models will need to

balance various aspects of predictive uncertainty, such as calibration and sharpness. This raises the question: what predictive uncertainty estimates work best for causal effect estimation using propensity scoring?

## 3.1 CALIBRATION: A NECESSARY CONDITION FOR PROPENSITY SCORING MODEL

This paper argues that calibration improves propensity-scoring methods. Intuitively, if the model $Q(T = 1|X)$ predicts a treatment assignment probability of 80%, then 80% of these predictions should receive the treatment. If the prediction is larger or smaller, the downstream IPTW estimator will overcorrect or undercorrect for the biased treatment allocation; see below for a simple example.

In other words, calibration is a *necessary condition* for a correct propensity scoring model. We formalize this intuition below, and we provide examples in Appendix H.2 where an IPTW estimator fails when it is not calibrated.

**Theorem 3.1.** *When $Q(T|X)$ is not calibrated, there exists an outcome function such that an IPTW estimator based on $Q$ yields an incorrect estimate of the true causal effect almost surely.*

*Example.* Consider $\mathcal{X} = \mathcal{T} = \mathcal{Y} = \{0,1\}$. Let $P(T = 1|X = 0) = p_0, P(T = 1|X = 1) = p_1$ and $P(X = 1) = 0.5$. Let us assume that $Q(T = 1|X = 0) = q_0$ and $Q(T = 1|X = 1) = q_1$. When $Q(T|X)$ is uncalibrated, $\exists i \in \{0,1\}, p_i \neq q_i$.

If $p_1 \neq q_1$, we set $Y = X \oplus T$ ($\oplus$ is logical 'AND'), and the IPTW estimator based on $Q$ obtains $\tau' = \frac{0.5.p_1}{q_1}$. Here, true ATE $\tau = 0.5$.

If $p_0 \neq q_0$, we set $Y = \bar{X} \oplus \bar{T}$ ($\bar{V}$ denotes logical negation of binary variable $V$), and the IPTW estimator based on $Q$ obtains $\tau' = \frac{-0.5(1-p_0)}{1-q_0}$. Here true ATE $\tau = -0.5$.

Please note that we require the model $Q$ to be uncalibrated and not necessarily inconsistent. $\square$

Please refer to Appendix H.2 for a full proof. Appendix H.4 also proves the following theorem for the AIPW estimator.

**Theorem 3.2.** *When propensity model $Q(T|X)$ is not calibrated and the outcome model f(X, T) is inaccurate for $X \in \{X : Q(T = 1|X) = q\} \subseteq \mathcal{X}$ such that $P(T = 1|Q(T = 1|X') = q) \neq q$, then there exists a true outcome function such that the doubly robust AIPW estimator based on Q and f yields an incorrect estimate of true causal effects almost surely.*

Thus, for the AIPW estimator, calibration is a necessary condition when the outcome model is inaccurate.

## 3.2 CALIBRATED UNCERTAINTIES IMPROVE PROPENSITY SCORING MODELS

In addition to being a necessary condition, we also identify settings in which calibration is either sufficient or prevents common failure modes of IPTW estimators. Specifically, we identify and study two such regimes: (1) accurate but over-confident propensity scoring models (e.g., neural networks [Guo et al., 2017]); (2) high-variance IPTW estimators that take as input numerically small propensity scores.

### 3.2.1 Error Bound on Causal Effect Estimates

Our first step for studying the role of calibration is to relate the error of an IPTW estimator to the difference between a model $Q(T|X)$ and the true $P(T|X)$. We define $\pi_{t,y}(Q) = \sum_x P(y|x,t)\frac{P(t|x)}{Q(t|x)}P(x)$ to be the estimated probability of $y$ given $t$ with a propensity score model $Q$. It is not hard to show that the true $Y[t] := \mathbb{E}_X Y[X,t] = \mathbb{E}_X \mathbb{E}[Y|X = x, \mathrm{do}(T = t)]$ can be written as $\sum_y y\pi_{y,t}(P)$ (see Appendix H.3). Similarly, the estimate of an IPTW estimator with propensity model $Q$ in the limit of infinite data tends to $\hat{Y}_Q[1] - \hat{Y}_Q[0]$, where $\hat{Y}_Q[t] := \sum_y y\pi_{y,t}(Q)$. We may bound the expected L1 ATE error $|Y[1] - Y[0] - (\hat{Y}_Q[1] - \hat{Y}_Q[0])|$ by $\sum_t |Y[t] - \hat{Y}_Q[t]| \leq \sum_t \sum_y |y| \cdot |\pi_{y,t}(P) - \pi_{y,t}(Q)|$.

Our first lemma bounds the error $|\pi_{y,t}(P) - \pi_{y,t}(Q)|$ as a function of the difference between $Q(T|X)$ and the true $P(T|X)$. A bound on the ATE error follows as a simple corollary.

**Lemma 3.3.** *The expected error $|\pi_{y,t}(P) - \pi_{y,t}(Q)|$ induced by an IPTW estimator with propensity score model $Q$ is bounded as*

$$|\pi_{y,t}(P) - \pi_{y,t}(Q)| \leq \mathbb{E}_{X \sim R_{y,t}}[\ell_\chi(P_t, Q_t)^{\frac{1}{2}}], \quad (4)$$

*where $R_{y,t} \propto P(Y = y|X, T = t)P(X)$ is a data distribution and $\ell_\chi(P_t, Q_t) = \left(1 - \frac{P(T=t|X)}{Q(T=t|X)}\right)^2$ is the chi-squared loss between the true propensity score and the model $Q$.*

*Proof (Sketch).* Note that $|\pi_{y,t}(P) - \pi_{y,t}(Q)| \leq \mathbb{E}_{X \sim R_{y,t}} \left|1 - \frac{P(T=t|X)}{Q(T=t|X)}\right| \leq \mathbb{E}_{R_{y,t}} \ell_\chi(P_t, Q_t)^{\frac{1}{2}}$ $\square$

See Appendix H.3.1 for the full proof.

**Corollary 3.4.** *Let $|y| \leq K$ for all $y \in \mathcal{Y}$. The error of an IPTW estimator with propensity score model $Q$ is bounded by $2|\mathcal{Y}|K \max_{y,t} \mathbb{E}_{R_{y,t}} \ell_\chi(P_t, Q_t)^{\frac{1}{2}}$.*

Note that $\ell_\chi$ is obtained from a proper scoring rule: it is small only if $Q$ correctly captures the probabilities in $P$.

A model that accurately outputs treatment assignment, but that does not output correct probability will have a large $\ell_\chi$; conversely, when $Q = P$, the bound equals to zero and the IPTW estimator is perfectly accurate. To the best of our knowledge, this is the first bound that relates the accuracy of an IPTW estimator directly to the quality of uncertainties of the probabilistic model $Q$. Corollary H.5 in Appendix H.4 obtains a similar upper bound on error of the doubly robust AIPW estimator that is proportional to the chi-squared loss $l_X$.

### 3.2.2 Calibration Reduces Variance of Estimators

A common failure mode of IPTW estimators arises when the probabilities from a propensity scoring model $Q(T|X)$ are small or even equal to zero—division by $Q(T|X)$ then causes the IPTW estimator to take on very large values or be undefined. Furthermore, when $Q(T|X)$ is small, small changes in its value cause large changes in the IPTW estimator, which induces problematically high variance. This failure mode also affects the doubly robust AIPW estimator, although it is more stable than the IPTW estimator.

Here, we show that calibration can help mitigate this failure mode. If $Q$ is calibrated, then it cannot take on abnormally small values relative to $P$. Specifically, if $P(T = t|X)$ is larger than some $\delta > 0$ such that $\delta < 1/2$, then any prediction from a calibrated estimate $Q$ of $P$ has to be larger than $\delta > 0$ as well. In other words, division by small numbers cannot be a greater problem than in the true model.

**Theorem 3.5.** *Let $P$ be the data distribution, and suppose that $1 - \delta > P(T|X) > \delta$ for all $T, X$ and let $Q$ be a calibrated model relative to $P$. Then $1 - \delta > Q(T|X) > \delta$ for all $T, X$ as well.*

*Proof (Sketch).* The proof is by contradiction. Suppose $Q(T = 1|x) = q$ for some $x$ and $q < \delta$. Then because $Q$ is calibrated, of the times when we predict $q$, we have $P(T = 1|Q(T = 1|X) = q) = q < \delta$, which is impossible since $P(T = 1|x) > \delta$ for every $x$.

See Appendix H.3.2 for the full proof. □

### 3.2.3 Calibration Improves Error Bounds

We show that calibration strictly improves our $\ell_\chi$ bound on the IPTW error.

**Theorem 3.6.** *Let $\ell_1$ be the expected bound on the error of an uncalibrated IPTW estimator $Q_1$ in Corollary 3.4, and let $\ell_2$ be the bound for $Q_2$, the recalibrated version of $Q_1$ with $\ell_\chi^{1/2}$ as the choice of loss $L$ to train the recalibrator. Then as the size of the calibration set $n \to \infty$ we have $\ell_2 \leq \ell_1$ with equality iff $Q_1 = Q_2$.*

---

**Algorithm 1** Calibrated Propensity Scoring

1. Split $\mathcal{D}$ into training set $\mathcal{D}'$ and calibration set $\mathcal{C}$
2. Train a propensity score model $Q(T|X)$ on $\mathcal{D}'$
3. Train recalibrator $R$ over output of $Q$ on $\mathcal{C}$
4. Apply IPW with $R \circ Q$ as prop. score model

---

*Proof (Sketch).* The part of $\ell_1, \ell_2$ that depends on $Q \in \{Q_1, Q_2\}$ is $L(Q, T) = \mathbb{E}_X \mathbb{E}_{T|X} \ell_\chi(Q(T = 1|X), T)^{1/2}$. In Section 4, we show that when we perform recalibration, it follows that $L(Q_2, T) = L(R \circ Q_1, T) \leq L(Q_1, T) + o(n)$ for a recalibrator $R$. As $n \to \infty$, $R \to B$, where B is a Bayes optimal recalibrator. If $Q_1 \neq Q_2$, then $L(Q_2, T) \neq L(Q_1, T)$ because $L$ is strictly proper. Conversely, when $Q_1 = Q_2$ clearly $\ell_1 = \ell_2$. Hence, the claim follows. □

Please refer to Appendix H.3.3 for a complete proof. Theorem H.6 in Appendix H.4 proves a similar result for the AIPW estimator when the outcome model is inaccurate.

### 3.2.4 Calibration and Accurate Causal Effect Estimation

If the model $Q$ is accurate enough to discriminate between different treatments (as might be the case with a powerful neural network), then calibration can ensure accurate IPTW estimates. This is a strong condition in practice. Please refer to Appendix H.3.4 for a detailed theoretical analysis.

Below, we also show that a post-hoc recalibrated model $Q'$ has vanishing regret $\ell(Q', Q)$ with respect to a base model $Q$ and a proper loss $\ell$ (including $\ell_\chi$ used in our calibration bound).

## 4 ALGORITHMS FOR CALIBRATED PROPENSITY SCORING

### 4.1 A FRAMEWORK FOR CALIBRATED PROPENSITY SCORING

Next, we propose algorithms that produce calibrated propensity scoring models. Our approach is outlined in Algorithm 1; it differs from standard propensity scoring methods by the addition of a post-hoc recalibration step (step #3) after training the model $Q$.

The recalibration step in Algorithm 1 implements a post-hoc recalibration procedure [Platt, 1999, Kuleshov et al., 2018] and is outlined in Algorithm 2. The key idea is to learn an auxiliary model $R : [0, 1] \to [0, 1]$ such that the joint model $R \circ Q$ is calibrated. Below, we argue that if $R$ can approximate the density $P(T = 1|Q(T|X) = p)$, $R \circ Q$ will be calibrated Kuleshov et al. [2018], Kuleshov and Deshpande [2022].

**Algorithm 2** Recalibration Step

**Input:** Pre-trained model $Q : \mathcal{X} \to [0,1]$, recalibrator $R : [0,1] \to [0,1]$, calibration set $\mathcal{C}$

**Output:** Recalibrated model $R \circ Q : \mathcal{X} \to [0,1]$.

1. Create a recalibrator training set:
   $$\mathcal{S} = \{(Q(x), y) \mid x, y \in \mathcal{C}\}$$

2. Fit the recalibration model $R$ on $\mathcal{S}$:
   $$\min_R \sum_{(p,y) \in \mathcal{S}} L(R(p), y)$$

---

Learning $R$ that approximates $P(T = 1 | Q(T|X) = p)$ requires specifying (1) a model class for $R$ and (2) a learning objective $\ell$. One possible model class for $R$ are non-parametric kernel density estimators over $[0,1]$; their main advantage is that they can provably learn the one-dimensional conditional density $P(T = 1 | Q(T|X) = p)$. Examples of such algorithms are RBF kernel density estimation or isotonic regression. Alternatively, one may use a family of parametric models for $R$: e.g., logistic regression, neural networks. Such parametric recalibrators can be implemented easily within deep learning frameworks and work well in practice, as we later demonstrate empirically.

Our learning objective for $R$ can be any proper scoring rule such as the L2 loss, the log-loss, or the Chi-squared loss. Optimizing it is a standard supervised learning problem.

## 4.2 ENSURING CALIBRATION IN PROPENSITY SCORING MODELS

Next, we seek to show that Algorithms 1 and 2 provably yield a calibrated model $R \circ Q$. This shows that the desirable property of calibration can be maintained in practice.

**Notation** We have a calibration dataset $\mathcal{C}$ of size $m$ sampled from $P$ and we train a recalibrator $R : [0,1] \to [0,1]$ over the outputs of a base model $Q$ to minimize a proper loss $L$. We denote the Bayes-optimal recalibrator by $B := P(T = 1 \mid Q(X))$; the probability of $T = 1$ conditioned on the forecast $(R \circ Q)(X)$ is $S := P(T = 1 \mid (R \circ Q)(X))$. To simplify notation, we omit the variable $X$, when taking expectations over $X, T$, e.g. $\mathbb{E}[L(R \circ Q, T)] = \mathbb{E}[L(R(Q(X)), T)]$.

Our first claim is that if we can perform density estimation, then we can ensure calibration. We first formally define the task of density estimation.

**Task 4.1** (Density Estimation). *The model $R$ approximates the density $B := P(T = t \mid Q(X))$. The expected proper loss of $R$ tends to that of $B$ as $m \to \infty$ such that w.h.p.:*

$$\mathbb{E}[L(B \circ Q, T)] \leq \mathbb{E}[L(R \circ Q, T)] < \mathbb{E}[L(B \circ Q, T)] + \delta$$

*where $\delta > 0$, $\delta = o(m^{-k}), k > 0$ is a bound that decreases with $m$.*

Note that non-parametric kernel density estimation is formally guaranteed to solve one-dimensional density estimation given enough data.

**Fact 4.2** (Wasserman [2004]). *When $R$ implements kernel density estimation and $L$ is the log-loss, Task 4.1 is solved with $\delta = o(1/m^{2/3})$.*

We now show that when we can solve Task 4.1, our approach yields models that are asymptotically calibrated in the sense that their calibration error tends to zero as $m \to \infty$.

**Theorem 4.3.** *The model $R \circ Q$ is asymptotically calibrated and the calibration error $\mathbb{E}[L_c(R \circ Q, S)] < \delta$ for $\delta = o(m^{-k}), k > 0$ w.h.p.*

See Appendix H.5.1 for the full proof.

## 4.3 NO-REGRET CALIBRATION

Next, we show that Algorithms 1 and 2 produce a model $R \circ Q$ that is asymptotically just as good as the original $Q$ as measured by the proper loss $L$.

**Theorem 4.4.** *The recalibrated model has asymptotically vanishing regret relative to the base model: $\mathbb{E}[L(R \circ Q, T)] \leq \mathbb{E}[L(Q, T)] + \delta$, where $\delta > 0, \delta = o(m)$.*

*Proof (Sketch).* Solving Task 4.1 implies $\mathbb{E}[L(R \circ Q, T)] \leq \mathbb{E}[L(B \circ Q, T)] + \delta \leq \mathbb{E}[L(Q, T)] + \delta$; the second inequality holds because a Bayes-optimal $B$ has lower loss than an identity mapping. □

See Appendix H.5.2 for the full proof. Thus, given enough data, we are guaranteed to produce calibrated forecasts and preserve base model performance as measured by $L$ (including $L_\chi$ used in our calibration bound).

# 5 EMPIRICAL EVALUATION

We perform experiments on several observational studies to evaluate calibrated propensity score models. We cover different types of treatment assignment mechanisms, base propensity score models, and varying dimensionality of observed covariates.

**Setup.** We use the Inverse-Propensity Treatment Weight (IPTW) and Augmented Inverse Propensity Weight (AIPW) estimators in our experiments. We compare the estimates obtained through calibrated propensities with several baselines including estimators based on uncalibrated propensity scores. We use sigmoid or isotonic regression as the recalibrator and utilize cross-validation splits to generate the calibration dataset (Appendix C). We measure the performance in terms of the absolute error in estimating ATE as $\epsilon_{ATE} = |\hat{\tau} - \tau|$, where $\tau$ is the true treatment effect and $\hat{\tau}$ is our estimated treatment effect.

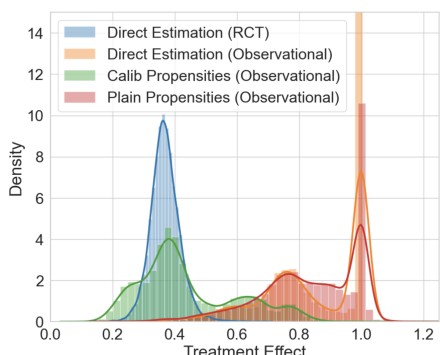

Figure 1: Recalibrating Propensity Score Model Reduces the Bias in Estimating Treatment Effect from Observational Data.

**Analysis of Calibration.** We evaluate the calibration of the propensity score model using expected calibration error (ECE), defined as $\mathbb{E}_{p \sim Q(T=1|X)}[|P(T=1|Q(T=1|X)=p)-p|]$, where $Q(T=1|X)$ models the treatment assignment mechanism $P(T=1|X)$. To compute ECE, we divide the probabilistic output range $[0,1]$ into equal-sized intervals $\{I_0, I_1, .., I_M\}$ such that we can generate buckets $\{B_i\}_{i=1}^{M}$, where $B_i = \{(X,T,Y)|Q(T=1|X) \in I_i\}$. The estimated ECE is then computed as

$$\text{ECE} = \sum_{i=1}^{M} \frac{|B_i|}{|\bigcup_{j=1}^{M} B_j|} |\text{avg}_i(B_i) - \text{pred}_i(B_i)|,$$

where $\text{avg}_i(B_i) = \sum_{j=1}^{|B_i|} T_j/|B_i|$ and $\text{pred}_i(B_i) = \sum_{j=1}^{|B_i|} Q(T=1|X_j)/|B_i|$.

### 5.1 DRUG EFFECTIVENESS STUDY

We simulate an observational study of recovery time from disease in response to the administration of a drug [Wilhelm, 2017]. The decision to treat an individual with the drug is dependent on the covariates specified as age, gender, and severity of disease. We use logistic regression as the propensity score model. In Figure 1, we see that weighing using recalibrated propensities allows us to approximate the distribution of individual treatment effect estimates better than uncalibrated propensities. Here, treatment effect estimates $\tau$ are computed as ratio $\mathbb{E}[Y[x,1]]/\mathbb{E}[Y[x,0]]$. The true average causal effect is 0.368. Please refer to Appendix D for details on the simulation, models used, and calibration plots.

In Table 1, we employ different treatment assignment mechanisms in each simulated observational study, allowing us to compare mechanisms that may or may not be well-specified by a linear model, e.g., Simulation C uses the logical AND condition while Simulation D uses the logical XOR condition to assign treatment. (Appendix D). We see that calibrated propensities produce lower absolute

error in estimating average treatment effect ($\epsilon_{ATE}$) under varying mechanisms. Here, the naive estimation computes the outcomes without weighing the samples with propensities. Uncertainty-calibrated propensities reduce bias more consistently as compared to weighing with plain propensity scores, propensity trimming [Lee et al., 2011], stabilized weights [Xu et al., 2010] and regularized covariate balancing optimization of propensity weights [Tan, 2019]. Since the optimal level of trimming is difficult to determine, it can sometimes increase bias, as seen in Simulation A. Similarly, the design of balancing equations impacts bias reduction in the covariate balancing approach [Ben-Michael et al., 2021] to calibration. We use the RCAL package [Tan and Sun, 2020] to implement the covariate balancing baseline. Table 6 in Appendix G reports the PEHE (Precision in Estimation of Heterogenous Effect) metric for all the experimental settings in Table 1 and demonstrates similar improvement with calibrated propensities. Table 7 in Appendix G demonstrates the effectiveness of calibration over six different base propensity models (including logistic regression) that approximate a fixed treatment assignment function.

In summary, calibrated propensities approximate the true distribution of individual treatment effects better and reduce the occurrence of numerically low scores. They reduce the error in ATE estimation across different propensity score models and treatment assignment mechanisms. In real-world observational studies, where we don't know the true treatment assignment mechanism, calibration can be useful to improve the treatment effect estimates from a potentially misspecified model.

### 5.2 UNSTRUCTURED IMAGE COVARIATES

We simulate a simple observational study following Louizos et al. [2017] and Deshpande et al. [2022] such that variables $X, T, Y \sim \mathbb{P}$ are binary and the true ATE is zero. Appendix E contains a detailed description of this simulation. We also introduce an unstructured image covariate $\mathbf{X}$ that represents $X$ as a randomly chosen MNIST image of a zero or one, depending on whether $X=0$ or $X=1$. Specifically, $\mathbb{P}(\mathbf{X}|X=1)$ is uniform over MNIST images of '1' and $\mathbb{P}(\mathbf{X}|X=0)$ is uniform over MNIST images of '0'.

We use a multi-layer perceptron as the propensity score model and recalibrate its output. In Table 2, we compare the IPTW estimates for ATE using binary $X$ and image $\mathbf{X}$ covariates (with $28 \times 28 = 784$ dimensions). The ECE is higher for the plain propensity score model trained on image covariates, indicating higher miscalibration with increasing covariate dimensions. We see that recalibration improves ATE estimates on high-dimensional image covariates.

Table 1: Recalibrating the Output of Propensity Score Model Results in Lower Error in Estimating Causal Effects. Reduction in ECE ($\Delta(ECE)$) implies that the calibration of the model improves with our technique. Results consisting of $\varepsilon_{ATE}$ are averaged over 10 experimental repetitions and braces contain the standard error.

| Setting | Sim A | Sim B | Sim C | Sim D |
|---|---|---|---|---|
| Naive | 0.498 (0.003) | 0.223 (0.003) | 0.279 (0.004) | 0.280 (0.006) |
| Plain propensities | 0.348 (0.035) | 0.211 (0.002) | 0.164 (0.002) | 0.075 (0.004) |
| Trimmed [Lee et al., 2011] | 0.481 (0.004) | 0.210 (0.002) | 0.153 (0.002) | 0.074 (0.004) |
| Stablized Wt [Xu et al., 2010] | 0.422 (0.016) | 0.210 (0.002) | 0.158 (0.003) | 0.078 (0.005) |
| Covariate Balancing [Tan, 2019] | 0.509 (0.003) | **0.190 (0.002)** | 0.169 (0.007) | 0.092 (0.013) |
| Calibrated (Ours) | 0.107 (0.029) | 0.195 (0.002) | **0.148 (0.001)** | **0.048 (0.010)** |
| Calibrated + Trimmed | 0.115 (0.028) | **0.193 (0.002)** | **0.148 (0.001)** | **0.048 (0.010)** |
| Calibrated + Stablized Wt | **0.057 (0.026)** | 0.194 (0.002) | **0.148 (0.001)** | **0.045 (0.009)** |
| $\Delta(ECE)$ | 0.010 (0.001) | 0.014 (0.001) | 0.025 (0.002) | 0.019 (0.001) |

Table 2: Reduction in ATE Estimation Error $\varepsilon_{ATE}$ with Structured and Unstructured Covariates.

| Setting | Naive Est. | Plain Propensities | Uncertainty Recalibration | $\Delta$(ECE) |
|---|---|---|---|---|
| Image Covariate | 0.187 (0.010) | 0.107 (0.029) | 0.095 (0.005) | 0.137 (0.046) |
| Binary Covariate | 0.176 (0.019) | 0.091 (0.011) | 0.085 (0.008) | 0.112 (0.029) |

## 5.3 GENOME-WIDE ASSOCIATION STUDIES

Genome-Wide Association Studies (GWASs) attempt to estimate the treatment effect of genetic mutations (called SNPs) on individual traits (called phenotypes) from observational datasets. Each SNP acts as a treatment. Confounding occurs because of hidden ancestry: individuals with shared ancestry have correlated genes and phenotypes.

The key takeaways can be summarized as follows. First, *recalibration enables off-the-shelf IPTW and AIPW estimators to match or outperform a state-of-the-art GWAS analysis system* (LMM/LIMIX; see Tables 3 and 4). Second, our method *enables the use of propensity score models that would otherwise be unusable* due to the poor quality of their uncertainty estimates (e.g., Naive Bayes; see Table 5). Third, leveraging new types of propensity score models that are fast to train (such as Naive Bayes), **improves the speed of GWAS analysis by more than two-fold** (see Table 5).

**Setup** We simulate the genotypes and phenotypes of individuals following a range of standard models as described in Appendix F. The outcome is simulated as $Y = \beta^T G + \alpha^T Z + \epsilon$, where $G$ is the vector of SNPs, $Z$ contains the hidden confounding variables, $\epsilon$ is noise distributed as Gaussian, $\beta$ is the vector of treatment effects corresponding to each SNP and $\alpha$ holds coefficients for the hidden confounding variables. We assume that the aspect of hidden population structure in $Z$ that needs to be controlled for is fully contained in the observed genetic data to ensure ignorability [Lin and Zeng, 2011]. To estimate the average marginal treatment effect corresponding to each SNP, we

iterate successively over the vector of SNPs such that the selected SNP is treatment $T$ and all the remaining SNPs are covariates $X$ for predicting the phenotypic outcome $Y$. The outcome is a vector of estimated treatment effects $\hat{\beta}$ corresponding to the vector of SNPs. We measure $\varepsilon_{ATE}$ as the $l_2$ norm of the difference between true and estimated marginal treatment effect vectors.

We use calibrated propensity scores with the IPTW and AIPW estimators to compute these treatment effects. We compare the performance of these estimators with standard methods to perform GWAS, including Principal Components Analysis (PCA) [Price et al., 2006, 2010], Factor Analysis (FA), and Linear Mixed Models (LMMs) [Yu et al., 2006, Lippert et al., 2011], implemented in the popular LIMIX library [Lippert et al., 2014]. Unless mentioned otherwise, 1% of total SNPs are causal and we have 4000 individuals in the dataset.

In Table 3, we demonstrate the effectiveness of estimators using calibrated propensities on five different GWAS datasets (Appendix F). Here, we have a total of 100 SNPs. In Table 4, we increase the proportion of causal SNPs for the Spatial simulation and continue to see improved performance under calibration. In Table 9 (Appendix F), we compare five different base models to learn propensity scores over six standard GWAS simulations and show that calibration improves the performance in each case. The performance of plain Naive Bayes as the base propensity score model is very poor owing to the simplistic conditional independence assumptions, but calibration improves its performance significantly. In Table 5, we compare the

Table 3: GWAS with Calibrated Propensities. We compare IPTW and AIPW estimates using calibrated propensity scores against standard baselines and a specialized GWAS analysis system (LMM/LIMIX).

| Dataset | Spatial ($\alpha$=0.1) | Spatial ($\alpha$=0.3) | Spatial ($\alpha$=0.5) | HGDP | TGP |
|---|---|---|---|---|---|
| Naive | 16.23 (0.91) | 11.76 (0.84) | 9.81 (0.69) | 11.82 (0.11) | 12.24 (0.71) |
| PCA | 9.60 (0.37) | 9.54 (0.41) | 9.38 (0.38) | 11.69 (0.20) | 10.73 (0.38) |
| FA | 9.55 (0.34) | 9.53 (0.44) | 9.23 (0.30) | 11.65 (0.16) | 10.59 (0.32) |
| LMM | 10.24 (0.41) | 9.58 (0.45) | **8.15 (0.40)** | **10.09 (0.35)** | **9.44 (0.57)** |
| IPTW (Calib) | **8.13 (0.35)** | **8.69 (0.56)** | 8.32 (0.34) | 10.86 (0.13) | **9.57 (0.58)** |
| IPTW (Plain) | 12.56 (1.25) | 10.22 (0.81) | 9.09 (0.48) | 11.62 (0.12) | 11.76 (0.86) |
| AIPW (Calib) | 8.94 (0.29) | 9.00 (0.58) | 8.59 (0.39) | 11.06 (0.12) | 10.32 (0.43) |
| AIPW (Plain) | 13.89 (0.76) | 10.46 (0.72) | 8.99 (0.51) | 11.38 (0.11) | 11.56 (0.65) |
| $\Delta_{ECE}$ | 0.022 (0.001) | 0.016 (0.007) | 0.015 (0.001) | 0.011 (0.001) | 0.022 (0.001) |

Table 4: Increasing Proportion of Causal SNPs. Calibrated propensities reduce the bias in treatment effect estimation across all setups and compare favorably against standard GWAS methods.

| Method | 1% Causal SNPs | 2% Causal SNPs | 5% Causal SNPs | 10% Causal SNPs |
|---|---|---|---|---|
| Naive | 22.408 (5.752) | 15.150 (2.213) | 23.388 (5.021) | 14.846 ( 2.272) |
| PCA | 18.104 (5.378) | 13.699 (2.413) | 15.837 (3.331) | 11.683 (0.983) |
| FA | 18.532 (3.641) | 14.166 (2.259) | 16.855 (2.764) | 11.963 (0.958) |
| LMM | 17.575 (3.408) | 13.896 (2.152) | 14.681 (3.366) | 10.108 (0.827) |
| IPTW (Calib) | **17.237 (3.054)** | **13.113 (1.775)** | **14.587 (3.432)** | **8.625 (0.838)** |
| IPTW (Plain) | 19.297 (3.425) | 14.372 (1.482) | 18.290 (3.788) | 11.859 (0.95240) |
| AIPW (Calib) | 17.647 (3.208) | 13.382 (1.676) | 15.166 (3.597) | 9.078 (0.928) |
| AIPW (Plain) | 20.652 (3.286) | 13.720 (1.798) | 21.321 (4.750) | 12.904 (1.990) |

computational throughput of calibrated Naive Bayes as the propensity score model with logistic regression. Here, we have a total of 1000 SNPs. We see that using calibrated Naive Bayes obtains performance competitive with logistic regression at a significantly higher throughput. Please refer to Appendix G for results on additional GWAS datasets.

Table 5: Calibrated Naive Bayes Yields Lower $\epsilon_{ATE}$ (IPTW) and Uses Lower Computational Resources As Compared to Logistic Regression.

| Method | $\epsilon_{ATE}$ | Tput (SNPs/sec) |
|---|---|---|
| LMM | 19.908 (3.592) | - |
| Calibrated NB | **18.210 (1.705)** | 47.6 |
| Plain NB | 1455.992 (185.084) | 68.6 |
| Calibrated LR | 23.618 (3.832) | 19.5 |
| Plain LR | 27.921 (4.713) | 20.1 |

## 6 RELATED WORK

Isotonic regression [Niculescu-Mizil and Caruana, 2005] and Platt scaling [Platt, 1999] are used to calibrate uncertainties over discrete outputs. This concept has been extended to regression calibration [Kuleshov et al., 2018] and online calibration [Kuleshov and Ermon, 2017]. Calibrated uncertainties have been used to improve deep reinforcement learning [Malik et al., 2019, Kuleshov and Deshpande, 2022], Bayesian optimization [Deshpande and Kuleshov, 2023], etc.

Lenis et al. [2018] and Kang and Schafer [2007] demonstrate the degradation in treatment effect estimation due to misspecified treatment and outcome models. Various modifications of propensity scores weights and different notions of calibration have been proposed to reduce the bias in treatment effect estimation [Imai and Ratkovic, 2014, Zhao, 2017, Ning et al., 2018, Van Der Laan et al., 2023, Stürmer et al., 2007b, Crump et al., 2009, Fan Li and Zaslavsky, 2018, Xu et al., 2010]. Appendix A compares our work with these approaches in more details.

As compared to covariate balancing calibration [Imai and Ratkovic, 2014] that modifies the underlying optimization procedure for obtaining balancing weights, our notion of calibration is simpler to implement and does not modify the optimization of the propensity score model. Unlike techniques like propensity weight trimming [Crump et al., 2009], our method does not introduce bias from throwing away weights beyond a pre-selected threshold.

Van Der Laan et al. [2023] and Xu and Yadlowsky [2022] apply the following notion of calibration for hetereogenous treatment effect (HTE) estimation: The average HTE of units with a given predicted HTE is equal to the shared predicted value. The goal of causal isotonic regression [Van Der Laan et al., 2023] is to ensure more directly that the predicted HTE outcome is reliable for different sub-groups of the population. Our work, on the other hand, calibrates the uncertainty outcome of the propensity score model that weighs the treated and control outcomes to achieve covariate balance. Although both calibration methods can be implemented using isotonic regression, the calibration guarantees are different. Our definition ensures that we avoid extreme propensity weights while balancing covariates and improve the error bounds on causal effect estimates. Our approach to calibration is applicable to HTE estimation (Appendix G, Table 6) and can be used with mis-specified propensity models that produce extreme weights. Applying our method to calibrate propensity scores in HTE estimation could be an interesting way to reduce the issue with extreme propensity weights while performing causal isotonic regression [Van Der Laan et al., 2023].

Uncertainty calibration can be combined independently with other methods like trimming [Crump et al., 2009, Fan Li and Zaslavsky, 2018], stabilized weights [Xu et al., 2010], covariate balancing techniques [Hainmueller, 2012, Chan et al., 2016, Zhao, 2017, Zubizarreta, 2015, Ning et al., 2018], etc. to improve the quality of ATE estimates.

# 7 DISCUSSION AND CONCLUSIONS

True treatment assignment mechanisms in observational studies are rarely known. Mis-specified propensity score models and outcome models may lead to biased treatment effect estimation [Kang and Schafer, 2007, Lenis et al., 2018]. We proposed a simple technique to perform post-hoc calibration of the propensity score model. We show that calibration is a necessary condition to obtain accurate treatment effects and calibrated uncertainties improve propensity scoring models. Empirically, we show that our technique reduces bias in estimates across a range of treatment assignment functions and base propensity score models. Propensity score models over high-dimensional, unstructured covariates like images, text, and genomic sequences are harder to specify, and we show that we can improve treatment effect estimates for such covariates over a range of base models including the popular logistic regression. We can calibrate simpler models like Naive Bayes over high-dimensional covariates and obtain higher computational throughput while maintaining competitive performance as measured by the error in treatment effect estimation.

**Limitations of Calibrated Propensities.** Calibration can ensure accurate causal effect estimates when the propensity score model Q can discriminate between different treatments. For example, if the propensity model outputs the marginal treatment distribution, i.e., $Q(T|X) = P(T)$, then $Q$ is perfectly calibrated but cannot estimate accurate treatment effects. Ensuring that Q can discriminate between different treatments is a strong condition and we discuss this further in Appendix H.3.4. When we use calibrated propensity scores for causal effect estimation, we assume that the observed covariates contain information on all the confounders. In the presence of unobserved confounders that cannot be recovered, calibrating the propensity scores will not be helpful.

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

# Calibrated Propensity Scores for Causal Effect Estimation (Supplementary Material)

**Shachi Deshpande**[1]                    **Volodymyr Kuleshov**[1]

[1]Dept of Computer Science, Cornell University and Cornell Tech, New York, NY, USA

## A    COMPARING UNCERTAINTY CALIBRATION WITH OTHER NOTIONS OF CALIBRATION FOR CAUSAL EFFECT ESTIMATION

Since true propensity model is often unknown in observational studies, the model we use to learn it is likely misspecified. Different parametric and non-parametric models have been proposed to learn propensity scores [McCaffrey et al., 2004, Hirano et al., 2003, Imbens, 2004, Lee et al., 2010]. Various strategies have been proposed to improve (and calibrate) the propensity score weights.

**Trimming and Overlapping Weights.**    When using inverse propensity score weights in causal effect estimators, small deviations in propensity score values can cause large errors in treatment effect estimation. Hence, several strategies have been proposed to trim the extreme propensity score weights [Crump et al., 2009, Li et al., 2018, Fan Li and Zaslavsky, 2018]. Trimming is known to introduce bias in causal effect estimates as we may lose the information on the magnitude of propensity scores from units that correspond to differences in covariate distributions. It is hard to determine the optimal trimming threshold upfront without sufficient knowledge of the observational study. This problem becomes more pronounced as we increase the complexity of the problem (e.g., multiple treatments [Lopez and Gutman, 2017]). Calibration, on the other hand, does not throw away the information contained within propensity scores weights below an arbitrarily chosen threshold. At the same time, it ensures that we do not produce propensity scores lower than the true propensity score (Theorem 3.5). Overlapping weights avoid extreme propensity weights by modifying the target population to include units that are more likely to obtain either of the binary treatments[Fan Li and Zaslavsky, 2018], while uncertainty calibrated propensities do not need to modify the target population.

**High-dimensional Covariates.**    Propensity score models also become unstable and show high variance when covariates are high-dimensional. When performing a direct adjustment of confounding without propensity scores, the estimation problem becomes more complex as the number of covariates increases (e.g., insufficient number of units to estimate outcome reliably for each combination of covariates). In our experiments with genome-wide association studies, we show that simple propensity models can be used for causal effect estimation with high-dimensional covariates through uncertainty calibration. Thus, calibrating propensities can allow us to estimate causal effects with simple (and potentially mis-specified) propensity score models when applying g-computation is infeasible due to high dimensionality.

**Covariate Balancing Calibration.**    Since the true propensity model is not known, researchers often modify and refit the propensity score model until covariate balance is achieved. Several techniques have been proposed to avoid this cyclic procedure and obtain covariate balance during optimization of the propensity weights  [Hainmueller, 2012, Imai and Ratkovic, 2014, Ning et al., 2018, Zhao, 2017, Zubizarreta, 2015, Chan et al., 2016]. Covariate balancing calibration is based on this idea and it solves an optimization problem such that we find weights that balance any averaged function of the covariates in treatment and control groups [Ben-Michael et al., 2021]. While these approaches show theoretical and empirical success in improving causal effect estimation, choices such as setting the appropriate balance conditions within the optimization problem require substantial knowledge of the observational study. Weights from the true propensity score are a solution to these balancing conditions. However, designing appropriate covariate balancing conditions becomes

harder as the dimensionality of the covariates increases. This is more challenging in the presence of covariate interactions (e.g., certain combinations of covariates representing socio-economic variables make an individual more likely to take up smoking as a treatment variable) and continuous covariates [Ben-Michael et al., 2021]. Uncertainty calibration of a potentially misspecified propensity score model does not change the base model optimization procedure and is simpler to implement on high-dimensional covariates. Thus, it can be more effective when we do not have enough information on the observational study to calibrate (optimize) using appropriate covariate balancing conditions.

**Causal Isotonic Calibration.** Van Der Laan et al. [2023] propose causal isotonic calibration to improve the estimation of heterogeneous treatment effects (HTEs). Their work enforces a different notion of calibration on the HTE prediction: The average HTE of units with a given predicted HTE is equal to the shared predicted value. The goal of their work is to ensure more directly that the predicted HTE outcome is reliable for different sub-groups of the population. Xu and Yadlowsky [2022] propose a technique to compute the calibration error while estimating heterogeneous treatment effects (HTEs) following this definition of calibration. Our work, on the other hand, calibrates the uncertainty outcome of the propensity score model that weighs the treated and control outcomes to achieve covariate balance. Our definition of calibration ensures that the number of units receiving treatment, given X % predicted probability of receiving treatment, is equal to X %. Although both calibration methods can be implemented using isotonic regression (with/without cross-validation splits to train the recalibrator), the calibration guarantees are different. Our definition ensures that we avoid extreme propensity weights while balancing covariates and improve the error bounds on causal effect estimates. Applying our method to calibrate propensity scores in HTE estimation could be an interesting way to reduce the issue with extreme propensity weights while performing causal isotonic regression [Van Der Laan et al., 2023] (e.g., in the case of high-dimensional/complex covariates). Although we only present results with the ATE metric in our main paper, our method can be applied to HTE estimation. Table 6 in Appendix G demonstrates that propensity score calibration also improves HTE estimation consistently in the drug effectiveness experiment from Table 1 in the main paper. It is also possible to apply our method independently to avoid extreme propensity scores when estimating the calibration error as proposed by Xu and Yadlowsky [2022].

**Other Ideas.** Other notions of propensity score calibration have been discussed in literature spanning survey sampling, missing data problems and causal inference [Lee and Valliant, 2009, Stürmer et al., 2007a]. However, these methods utilize a different setup (for example, access to validation dataset with information on extra variables) to perform calibration. Our method performs calibration under absence of hidden confounding and does not require accessing extra datasets (our calibration dataset can be generated with cross-validation).

# B   ESTIMATORS FOR AVERAGE TREATMENT EFFECTS

We expressed ATE as $\tau = \mathbb{E}\left(\frac{TY}{e(X)} - \frac{(1-T)Y}{1-e(X)}\right)$. Following Smith et al. [2020], we can simplify the following term

$$
\mathbb{E}\left[\frac{TY}{e(X)}\right] = \mathbb{E}[\mathbb{E}\left(\frac{TY}{e(X)}|T,X\right)]
$$

$$
= \mathbb{E}[\left(\frac{T\mathbb{E}(Y|T,X)}{e(X)}\right)]
$$

$$
= \mathbb{E}[\left(\frac{T\mathbb{E}(Y|T=1,X)}{e(X)}\right)]
$$

$$
= \mathbb{E}[\mathbb{E}\left(\frac{T\mathbb{E}(Y|T=1,X)}{e(X)}|X\right)]
$$

$$
= \mathbb{E}[\left(\frac{\mathbb{E}(Y|T=1,X)P(T=1|X)}{e(X)}\right)]
$$

$$
= \mathbb{E}[\mathbb{E}(Y|T=1,X)].
$$

Similarly,

$$
\mathbb{E}\left[\frac{(1-T)Y}{1-e(X)}\right] = \mathbb{E}[\mathbb{E}(Y|T=0,X)].
$$

Thus, we can show that ATE is indeed equivalent to $\mathbb{E}\left(\frac{TY}{e(X)} - \frac{(1-T)Y}{1-e(X)}\right)$.

Due to sensitivity of the IPTW estimator toward mis-specification of propensity score model, Robins et al. [1994] propose doubly robust Augmented Inverse Propensity Weighted (AIPW) estimator for ATE. The AIPW estimate is asymptotically unbiased when either the treatment assignment (propensity) model or the outcome model is well-specified.

We define the outcome model as $f(X = x, T = t)$ to approximate the outcome $Y[X = x, T = t]$ as defined in Section 2.

With this, we define the AIPW estimator as

$$\hat{\tau} = \frac{1}{n} \sum_{i=1}^{n} \left[ f(X_i, T = 1) - f(X_i, T = 0) + \frac{T_i(Y_i - f(X_i, T = 1))}{e(X_i)} - \frac{(1 - T_i)(Y_i - f(X_i, T = 0))}{1 - e(X_i)} \right]$$

## C  ADDITIONAL DETAILS ON THE CALIBRATION ALGORITHM

Algorithm 1 depends linearly on the number of data-splits created (training set and calibration set) in addition to the time-complexity of training the propensity model $Q(T|X)$ and recalibrator (Algorithm 2). The time complexity will also depend on an additive term corresponding to computing $R \circ Q$ for all data-points in dataset $\mathcal{D}$. Space complexity depends linearly on the size of dataset $\mathcal{D}$ together with additive terms for model size of $Q(T|X)$ and $R$.

**Designing the Recalibration Method.**  When the treatments are binary, we can choose between isotonic regression and logistic regression as the recalibrator. Since isotonic regression is prone to overfitting, we prefer to use logistic regression when the calibration dataset size is small (e.g., <1000 data points). Leave-one-out cross-validation splits could be useful to generate the calibration dataset when the dataset size is small. When moving to the multiple treatment/ continuous treatment setup, designing the recalibrator may involve more choices (for example, we can have a simple neural network as a recalibrator in the case of continuous treatments). Using a separate cross-validation dataset would help select these hyperparameters.

**Cross-validation Splits.**  The requirement to allocate a separate calibration dataset may reduce the size of dataset available for training the propensity score model $Q(T|X)$. Hence, we can use cross-validation splits in the dataset to calibrate a propensity score model. To implement this approach, we divide our dataset D into $k$ partitions $S_1, S_2, .., S_k$. For each dataset split $S_k$, we train the propensity score model $Q_k(T|X)$ on $S_k$ and and generate parts of recalibrator training dataset (as defined in Algorithm 2) as $C_k = \{Q_k(x), y | x, y \in D - S_k\}$. After this, we can take a union over all $C_k$ to generate the complete recalibrator training dataset. This allows us to use the entire available dataset for training the propensity score model as well as the recalibrator. This can be useful especially when the available dataset size is small. In our experiments, we have used leave-one-out cross-validation splits (thus, each partition $S_k$ is of size n-1 where n is the size of dataset D).

## D  DRUG EFFECTIVENESS SIMULATIONS

The covariates contain gender ($x_1$), age ($x_2$) and disease severity ($x_3$), while treatment ($t$) corresponds to administration of drug. Outcome ($y$) is the time taken for recovery of patient.

We simulate the covariates as

$$x_1 \sim \text{Bernoulli}(0.5) \qquad x_2 \sim \text{Gamma}(\alpha = 8, \beta = 4) \qquad x_3 \sim \text{Beta}(\alpha = 3, \beta = 1.5).$$

The outcome is simulated as

$$y \sim \text{Poisson}(2 + 0.5x_1 + 0.03x_2 + 2x_3 - t).$$

The treatment $t$ is assigned on the basis of the covariates age, gender and severity of disease defined above. The simulations differ in their treatment assignment functions, which are described as follows

1. Simulation A: If ($x_1 = 1$), set $t = (x_2 > 45)$ else set $t = (x_3 > 0.3)$.
2. Simulation B: If ($x_1 = 1$), set $t = (x_3 > 0.3)$ else set $t = (x_2 > 40)$.
3. Simulation C: If $x_2 > 50$ AND $x_3 > 0.7$ then set $t = 1$ else $t = 0$.
4. Simulation D: If $x_2 > 50$ XOR $x_3 > 0.7$ then set $t = 1$ else $t = 0$.

For a linear model predicting treatment given covariates, Simulation C is easier to learn as compared to A, B and D.

Table 7 works with a slightly modified simulation D, where the treatment is set to 1 with probability of 0.99 when the XOR condition is true (otherwise 0), while it is set to 0 with probability 0.99 when the condition is false.

**Experimental Setup.** We model the outcome using random forests such that the covariates and treatment is taken as input. Logistic regression is used as the propensity score model and the inverse propensity scores are used to weigh each sample while training the outcome model. We use isotonic regression as the recalibrator. The treatment effect is expressed as the ratio $\mathbb{E}(Y(1))/\mathbb{E}(Y(0))$, where $Y(T)$ is the potential outcome $Y$ obtained by setting treatment to $T$. The outcome is time taken by the patient to make full recovery from the disease. We use 10 cross-val splits to generate the recalibration dataset.

The trimming baseline clips propensity weights to threshold of 0.001. Thresholds of 0.001-0.01 are applied commonly when using causal effect estimators based on inverse propensities.

The experiments were run on a laptop with 2.8GHz quad-core Intel i7 processor.

In Figure D, we see that the calibration curve of propensity score model gets closer to the diagonal after applying recalibration.

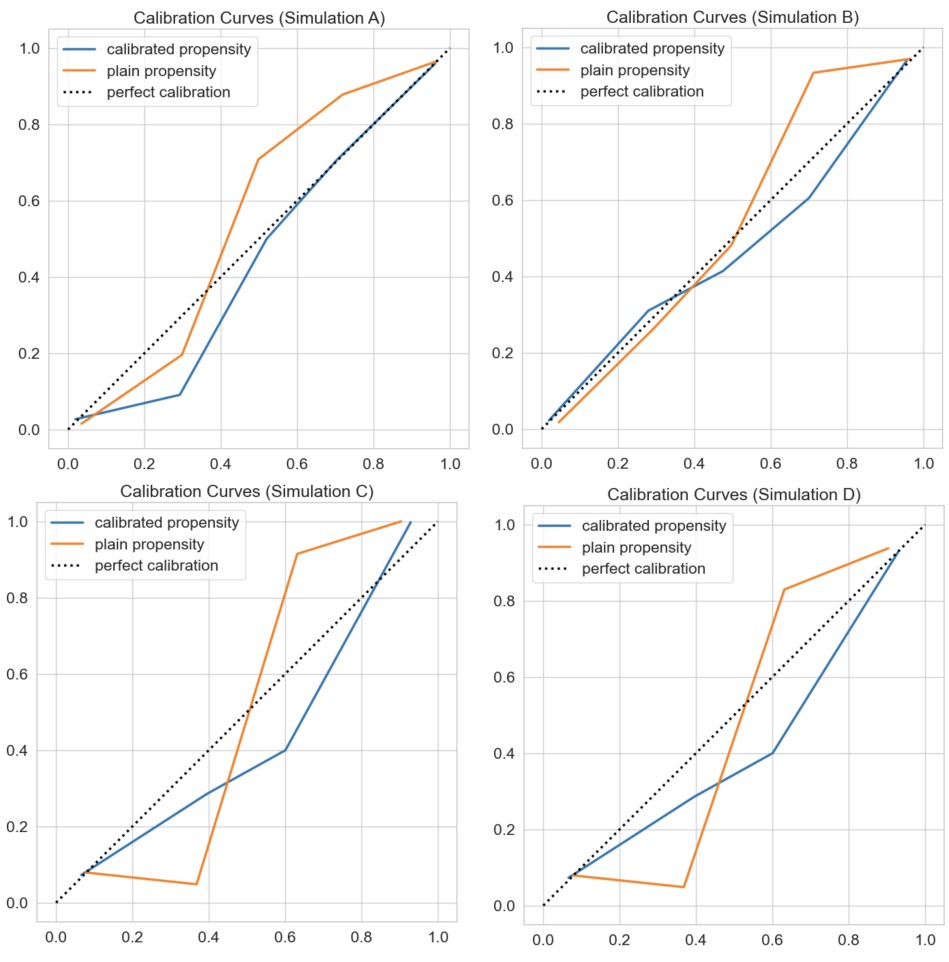

Figure 2: Calibration of propensity score model for Drug Effectiveness Study.

# E  UNSTRUCTURED COVARIATES EXPERIMENT

Following Louizos et al. [2017], we generate a synthetic observational dataset consisting of binary variables $X, T, Y \sim \mathbb{P}$, such that

$$\mathbb{P}(Z=1) = \mathbb{P}(Z=0) = 0.5 \qquad \mathbb{P}(X=1|Z=1) = 0.3 \qquad \mathbb{P}(X=1|Z=0) = 0.1$$
$$\mathbb{P}(T=1|Z=1) = 0.4 \qquad \mathbb{P}(T=1|Z=0) = 0.2 \qquad Y = T \oplus Z.$$

Louizos et al. [2017] show that the true ATE under this simulation is zero. The presented results include propensity weight trimming by threshold of 0.01.

The simulation generation as well as ATE estimation experiments were done on a laptop with 2.8GHz quad-core Intel i7 processor.

# F  SIMULATED GWAS DATASETS

We have $N$ individuals and $M$ number of total SNPs for each individual. Thus, we need to simulate a SNP matrix $G \in \{0,1\}^{N \times M}$ and an outcome vector $Y \in \mathbb{R}^N$. We also have a matrix of confounding variables $Z \in \mathbb{R}^{N \times D}$ for these $N$ individuals. We do not observe the confounding variables. Following Wang and Blei [2019], we generate the following genotype simulations.

To generate the SNP matrix, we generate an allele frequency matrix $F \in \mathbb{R}^{N \times M}$ such that $F = S\Gamma^{\top}$, where $S \in \mathbb{R}^{N \times D}$ encodes genetic population structure and $\Gamma \in \mathbb{R}^{M \times D}$ maps how structure affects alleles.

Thus, $g_{ij} \sim \text{Binomial}(1, F_{ij})$.

The outcome is modeled as $Y = \beta^T G + \alpha^T Z + \epsilon$, where $\beta$ is the vector of treatment effects for each SNP, $\alpha$ is the vector of coefficients corresponding to the hidden confounders in $Z$ and $\epsilon$ is noise distributed independently as a Gaussian.

We simulate a high signal-to-noise ratio while simulating outcomes by replacing $\lambda_i = (\alpha^T Z)_i$ as

$$\lambda_i \leftarrow \left[ \frac{s.d.\{\sum_{j=1}^m \beta_j g_{ij}\}_{i=1}^N}{\sqrt{\nu_{gene}}} \right] \left[ \frac{\sqrt{\nu_{conf}}}{s.d.\{\lambda_i\}_{i=1}^N} \right] \lambda_i$$

$$\epsilon_i \leftarrow \left[ \frac{s.d.\{\sum_{j=1}^m \beta_j g_{ij}\}_{i=1}^N}{\sqrt{\nu_{gene}}} \right] \left[ \frac{\sqrt{\nu_{noise}}}{s.d.\{\epsilon_i\}_{i=1}^n} \right] \epsilon_i,$$

where $\nu_{gene} = 0.4, \nu_{conf} = 0.4$, and $\nu_{noise} = 0.2$.

Below, we reproduce the simulation details as described by Wang and Blei [2019]. $\Gamma$ and $S$ are simulated in different ways to generate the following datasets.

1. **Spatial Dataset**: The matrix $\Gamma$ was generated by sampling $\gamma_{ik} \sim 0.9 \times \text{Uniform}(0, 0.5)$, for $k = 1, 2$ and setting $\gamma_{ik} = 0.05$. The first two rows of S correspond to coordinates for each individual on the unit square and were set to be independent and identically distributed samples from $\text{Beta}(\alpha, \alpha), \alpha = 0.1, 0.3, 0.5$, while the third row of $S$ was set to be 1, i.e. an intercept. As $\alpha \implies 0$, the individuals are placed closer to the corners of the unit square, while when $\alpha = 1$, the individuals are distributed uniformly.

2. **Balding-Nichols Model (BN)**: Each row i of $\Gamma$ has three independent and identically distributed draws taken from the Balding-Nichols model: $\gamma_{ik} \sim BN(p_i, F_i)$, where $k \in 1, 2, 3$. The pairs $(p_i, F_i)$ are computed by randomly selecting a SNP in the HapMap data set, calculating its observed allele frequency and estimating its FST value using the Weir & Cockerham estimator [Weir and Cockerham, 1984]. The columns of $S$ were Multinomial(60/210,60/210,90/210), which reflect the subpopulation proportions in the HapMap dataset.

3. **1000 Genomes Project (TGP)** [1000 Genomes Project Consortium et al., 2015]: The matrix $\Gamma$ was generated by sampling $\gamma_{ik} \sim 0.9\text{Uniform} \times (0, 0.5)$, for $k = 1, 2$ and setting $\gamma_{ik} = 0.05$. In order to generate $S$, we compute the first two principal components of the TGP genotype matrix after mean centering each SNP. We then transformed each principal component to be between (0,1) and set the first two rows of $S$ to be the transformed principal components. The third row of $S$ was set to 1, i.e. an intercept.

4. **Human Genome Diversity Project (HGDP)** [Fairley et al., 2019, Bergström et al., 2020]: Same as TGP but generating S with the HGDP genotype matrix.

These simulations and the ATE estimation experiments were all done on a laptop with 2.8GHz quad-core Intel i7 processor. The presented results include propensity weight trimming by threshold of 0.01 (after applying a possible calibration step).

# G    ADDITIONAL EXPERIMENTAL RESULTS

For the Drug Effectiveness simulations, Table 7, we compare a range of base propensity score models where the true treatment assignment function is non-linear logical XOR (Appendix D). We see the benefits of calibration across varying degrees of mis-specification in the base model. After calibration, non-linear MLP and SVM (RBF) show the best $\varepsilon_{ATE}$, while mis-specified linear models like logistic regression also show consistent reduction in $\varepsilon_{ATE}$. We observe a greater reduction in bias ($\varepsilon_{ATE}$) with lowering ECE.

Table 6 extends Table 1 with the PEHE metric on all the simulation settings.

For the GWAS experiments, we provide a complete table of dataset simulations and acomparison against different base propensity models in Table 8 and Table 9 respectively.

Table 6: Recalibrating the Output of Propensity Score Model Results in Lower Error in Estimating Causal Effects. Reduction in ECE ($\Delta(ECE)$) implies that the calibration of the model improves with our technique. Results consisting of PEHE are averaged over 10 experimental repetitions and braces contain the standard error.

| Setting | Sim A | Sim B | Sim C | Sim D |
|---|---|---|---|---|
| Naive | 0.263 (0.002) | 0.075 (0.002) | 0.105 (0.002) | 0.103 (0.003) |
| Plain propensities | 0.149 (0.024) | 0.068 (0.001) | 0.052 (0.001) | **0.031 (0.001)** |
| Trimmed  [Lee et al., 2011] | 0.245 (0.004) | 0.067 (0.001) | 0.046 (0.001) | **0.031 (0.001)** |
| Stablized Wt [Xu et al., 2010] | 0.195 (0.013) | 0.076 (0.002) | 0.114 (0.004) | 0.112 (0.005) |
| Covariate Balancing [Tan, 2019] | 0.280 (0.003) | **0.056 (0.001)** | 0.050 (0.003) | 0.107 (0.007) |
| Calibrated (Ours) | 0.047 (0.010) | **0.057 (0.001)** | **0.042 (0.001)** | **0.032 (0.001)** |
| Calibrated + Trimmed | 0.049 (0.010) | **0.057 (0.001)** | **0.042 (0.001)** | **0.032 (0.001)** |
| Calibrated + Stablized Wt | **0.030 (0.007)** | **0.057 (0.001)** | **0.042 (0.001)** | 0.033 (0.001) |
| $\Delta(ECE)$ | 0.010 (0.001) | 0.014 (0.001) | 0.025 (0.002) | 0.019 (0.001) |

Table 7: Comparison of different base propensity score models. (Sim D)

| Base model | $\varepsilon_{ATE}$(Plain) | ECE (Plain) | $\varepsilon_{ATE}$ (Calib) | ECE (Calib) |
|---|---|---|---|---|
| Log. Reg. | 0.031 (0.003) | 0.124 (0.001) | 0.016 (0.002) | 0.018 (0.001) |
| MLP | 0.014 (0.005) | 0.075 (0.002) | 0.008 (0.003) | 0.012 (0.002) |
| SVM (Linear) | 0.032 (0.005) | 0.126 (0.001) | 0.015 (0.003) | 0.017 (0.001) |
| SVM (RBF) | 0.012 (0.003) | 0.020 (0.000) | 0.009 (0.004) | 0.011 (0.001) |
| Adaboost | 0.039 (0.003) | 0.296 (0.001) | 0.022 (0.004) | 0.037 (0.008) |
| Naive Bayes | 0.022 (0.004) | 0.146 (0.001) | 0.017 (0.003) | 0.016 (0.002) |

Table 8: GWAS with calibrated propensities. We compare IPTW and AIPW estimates using calibrated propensity scores against several standard GWAS baselines. $\varepsilon_{ATE}$ is the $l_2$ norm of difference between true and estimated marginal treatment effect vector. Under all setups, calibrated propensities improve the treatment effect estimates.

| Dataset | Spatial ($\alpha$=0.1) | Spatial ($\alpha$=0.3) | Spatial ($\alpha$=0.5) | Balding Nichols | HGDP | TGP |
|---|---|---|---|---|---|---|
| Naive | 16.23 (0.91) | 11.76 (0.84) | 9.81 (0.69) | 19.25 (1.17) | 11.82 (0.11) | 12.24 (0.71) |
| PCA | 9.60 (0.37) | 9.54 (0.41) | 9.38 (0.38) | 14.12 (1.28) | 11.69 (0.20) | 10.73 (0.38) |
| FA | 9.55 (0.34) | 9.53 (0.44) | 9.23 (0.30) | **12.59 (1.05)** | 11.65 (0.16) | 10.59 (0.32) |
| LMM | 10.24 (0.41) | 9.58 (0.45) | **8.15 (0.40)** | **13.13 (1.09)** | **10.09 (0.35)** | **9.44 (0.57)** |
| IPTW (Calib) | **8.13 (0.35)** | **8.69 (0.56)** | 8.32 (0.34) | 13.62 (0.68) | 10.86 (0.13) | **9.57 (0.58)** |
| IPTW (Plain) | 12.56 (1.25) | 10.22 (0.81) | 9.09 (0.48) | 14.36 (0.74) | 11.62 (0.12) | 11.76 (0.86) |
| AIPW (Calib) | 8.94 (0.29) | 9.00 (0.58) | 8.59 (0.39) | 16.81 (1.39) | 11.06 (0.12) | 10.32 (0.43) |
| AIPW (Plain) | 13.89 (0.76) | 10.46 (0.72) | 8.99 (0.51) | 17.66 (1.33) | 11.38 (0.11) | 11.56 (0.65) |
| $\Delta_{ECE}$ | 0.022 (0.001) | 0.016 (0.007) | 0.015 (0.001) | 0.013 (0.002) | 0.011 (0.001) | 0.022 (0.001) |

Table 9: Comparing propensity score models. We compare the AIPW estimate using calibrated propensities and observe reduction in error across a range of base propensity score models.

| Dataset | Metrics | LR | MLP | Random Forest | Adaboost | NB |
|---|---|---|---|---|---|---|
| Spatial ($\alpha$=0.1) | $\varepsilon_{ATE}$ (plain) | 13.886 (0.755) | 17.403 (1.070) | 12.911 (0.612) | 16.234 (0.916) | 582.731 (64.514) |
| | $\varepsilon_{ATE}$ (calib) | 8.942 (0.287) | 14.661 (0.762) | 8.706 (0.322) | 8.524 (0.297) | 8.526 (0.472) |
| | $\Delta_{ECE}$ | 0.022 (0.001) | 0.072 (0.003) | 0.060 (0.001) | 0.252 (0.006) | 0.281 (0.002) |
| Spatial ($\alpha$=0.3) | $\varepsilon_{ATE}$ (plain) | 10.460 (0.720) | 12.636 (0.730) | 10.578 (0.768) | 11.764 (0.839) | 400.643 (49.301) |
| | $\varepsilon_{ATE}$ (calib) | 9.000 (0.58) | 11.550 (0.747) | 9.277 (0.532) | 8.909 (0.549) | 9.121 (0.535) |
| | $\Delta_{ECE}$ | 0.016 (0.007) | 0.070 (0.003) | 0.063 (0.001) | 0.244 (0.006) | 0.281 (0.002) |
| Spatial ($\alpha$=0.5) | $\varepsilon_{ATE}$ (plain) | 8.990 (0.510) | 10.408 (0.694) | 9.277 (0.518) | 9.814 (0.691) | 276.017 (24.183) |
| | $\varepsilon_{ATE}$ (calib) | 8.590 (0.390) | 9.728 (0.650) | 8.687 (0.224) | 8.520 (0.286) | 8.592 (0.216) |
| | $\Delta_{ECE}$ | 0.015 (0.001) | 0.070 (0.002) | 0.065 (0.001) | 0.239 (0.007) | 0.269 (0.003) |
| Balding Nichols | $\varepsilon_{ATE}$ (plain) | 17.660 (1.330) | 18.282 (1.267) | 18.419 (1.210) | 19.248 (1.169) | 95.892 (6.350) |
| | $\varepsilon_{ATE}$ (calib) | 16.810 (1.390) | 17.033 (1.391) | 16.611 (1.385) | 16.938 (1.367) | 16.833 (1.392) |
| | $\Delta_{ECE}$ | 0.013 (0.002) | 0.041 (0.002) | 0.052 (0.002) | 0.259 (0.010) | 0.261 (0.009) |
| HGDP | $\varepsilon_{ATE}$ (plain) | 11.380 (0.110) | 12.358 (0.197) | 11.529 (0.107) | 11.816 (0.108) | 138.086 (5.086) |
| | $\varepsilon_{ATE}$ (calib) | 11.060 (0.120) | 11.198 (0.106) | 11.299 (0.143) | 11.070 (0.123) | 11.430 (0.133) |
| | $\Delta_{ECE}$ | 0.011 (0.001) | 0.069 (0.002) | 0.053 (0.001) | 0.275 (0.006) | 0.206 (0.003) |
| TGP | $\varepsilon_{ATE}$ (plain) | 11.560 (0.650) | 11.965 (0.754) | 11.677 (0.614) | 12.246 (0.713) | 87.329 (5.716) |
| | $\varepsilon_{ATE}$ (calib) | 10.320 (0.430) | 11.530 (0.633) | 10.519 (0.402) | 10.244 (0.398) | 9.070 (0.316) |
| | $\Delta_{ECE}$ | 0.022 (0.001) | 0.061 (0.002) | 0.070 (0.002) | 0.204 (0.007) | 0.267 (0.004) |

# H  THEORETICAL ANALYSIS

## H.1  NOTATION

As described in Section 2, we are given an observational dataset $\mathcal{D} = \{(x^{(i)}, y^{(i)}, t^{(i)})\}_{i=1}^n$ consisting of $n$ units, each characterized by features $x^{(i)} \in \mathcal{X} \subseteq \mathbb{R}^d$, a binary treatment $t^{(i)} \in \{0, 1\}$, and a scalar outcome $y^{(i)} \in \mathcal{Y} \subseteq \mathbb{R}$. We assume $\mathcal{D}$ consists of i.i.d. realizations of random variables $X, Y, T \sim P$ from a data distribution $P$. Although we assume binary treatments and scalar outcomes, our approach naturally extends beyond this setting. The feature space $\mathcal{X}$ can be any continuous or discrete set.

## H.2 CALIBRATION: A NECESSARY CONDITION FOR PROPENSITY SCORING MODELS

**Theorem H.1.** *When $Q(T|X)$ is not calibrated, there exists an outcome function such that an IPTW estimator based on $Q$ yields an incorrect estimate of the true causal effect almost surely.*

*Example.* Consider a toy binary setting where $\mathcal{X} = \mathcal{T} = \{0, 1\}, \mathcal{Y} = \{0, 1\}^2$.

We set $Y = (X \oplus T, \bar{X} \oplus \bar{T})$, $P(T = 1|X = 0) = p_0, P(T = 1|X = 1) = p_1$ and $P(X = 1) = 0.5$ such that $\oplus$ is logical 'AND' and $\bar{V}$ denotes logical negation of binary variable $V$. We see that true ATE is $\tau = (0.5, -0.5)$. Let us assume that $Q(T = 1|X = 0) = q_0$ and $Q(T = 1|X = 1) = q_1$. Thus, with IPTW estimator based on $Q$, we estimate $\tau' = \mathbb{E}\left(\frac{TY}{Q(T=1|X)} - \frac{(1-T)Y}{1-Q(T=1|X)}\right) = (\frac{0.5.p_1}{q_1}, -\frac{0.5(1-p_0)}{1-q_0})$. The treatment effect $\tau' = \tau$ only when $q_0 = p_0$ and $q_1 = p_1$, which is not true if $Q$ is not calibrated. Although this example allows multidimensional outcomes, this shows that we can pick an outcome function such that uncalibrated model $Q$ produces inaccurate treatment effect estimates using the IPTW estimator. □

*Proof.* Let $\mathcal{P}$ be a space of valid probability distributions on $\mathcal{Y}$. We would like to prove that $\exists P'(Y|X = x, T = t) \in \mathcal{P}$ such that

$$\lim_{n \to \infty} \text{Probability}(\hat{\tau}_n = \tau) = 0$$

where

- $\tau$ is the true ATE

- $\hat{\tau}_n$ is the ATE estimated using IPTW estimator such that we have $n$ individuals and propensity score model is $Q(T = 1|X)$

- The probability is taken over all propensity models $Q(T = 1|X)$ such that $\exists q \in [0, 1], P(T = 1|Q(T = 1|X) = q) \neq q$, and all data-generating distributions $P'(Y, T, X) = P'(Y|X, T).P(T, X)$.

Let $S_Q = \{q | \exists X \in \mathcal{X}, Q(T = 1|X) = q\}$. We partition $\mathcal{X}$ into buckets $\{B_q\}_{q \in S_Q}$ such that $B_q = \{X | Q(T = 1|X) = q\}$.

Let $\hat{\tau}(Q) = \lim_{n \to \infty} \tau_n$. Thus, for discrete $\mathcal{X}$, we could write

$$\hat{\tau}(Q) = \mathbb{E}_{Y \sim P'(.|T,X); T, X \sim P}\left[\left(\frac{TY}{Q(T=1|X)} - \frac{(1-T)Y}{1-Q(T=1|X)}\right)\right]$$

Computing expectation over $X$

$$= \sum_{X \in \mathcal{X}} \mathbb{E}_{Y \sim P'(.|T,X); T \sim P(.|X)}\left[\left(\frac{TY}{Q(T=1|X)} - \frac{(1-T)Y}{1-Q(T=1|X)}\right) P(X)\right]$$

Computing expectation over $T$

$$= \sum_{X \in \mathcal{X}} \mathbb{E}_{Y \sim P'(.|X,T=1)}\left[\left(\frac{P(T=1|X)Y}{Q(T=1|X)}\right) P(X)\right] + \sum_{X \in \mathcal{X}} \mathbb{E}_{Y \sim P'(.|X,T=0)}\left[\left(-\frac{(1-P(T=1|X))Y}{1-Q(T=1|X)}\right) P(X)\right]$$

$$= \sum_{X \in \mathcal{X}} \left(\mathbb{E}_{Y \sim P'(.|X,T=1)}\left[\left(\frac{P(T=1|X)Y}{Q(T=1|X)}\right)\right] - \mathbb{E}_{Y \sim P'(.|X,T=0)}\left[\left(\frac{(1-P(T=1|X))Y}{1-Q(T=1|X)}\right)\right]\right) P(X)$$

Expressing the summation over $X$ differently

$$= \sum_{q \in S_Q} \sum_{X \in B_q} \left(\mathbb{E}_{Y \sim P'(.|X,T=1)}\left[\left(\frac{P(T=1|X)Y}{Q(T=1|X)}\right)\right] - \mathbb{E}_{Y \sim P'(.|X,T=0)}\left[\left(\frac{(1-P(T=1|X))Y}{1-Q(T=1|X)}\right)\right]\right) P(X)$$

Since $Q(T = 1|X)$ is not calibrated, we know that $\exists q \in [0, 1], P(T = 1|Q(T = 1|X) = q) \neq q$. Let us pick $q' \in S_Q$ such that $P(T = 1|Q(T = 1|X) = q') \neq q'$.

We could design $P'(Y|X, T) = \mathbb{I}(Y = T.\mathbb{I}(X \in B_{q'}))/P(X \in B_{q'})$.

Now, we can write

$$\hat{\tau}(Q) = \sum_{q \in S_Q} \sum_{X \in B_q} \left( \mathbb{E}_{Y \sim P'(.|X,T=1)} \left[ \left( \frac{P(T=1|X)Y}{Q(T=1|X)} \right) \right] - \mathbb{E}_{Y \sim P'(.|X,T=0)} \left[ \left( \frac{(1-P(T=1|X))Y}{1-Q(T=1|X)} \right) \right] \right) P(X)$$

(Since $Y = 0$ when $T = 0$ or $X \notin B_{q'}$)

$$= \sum_{X \in B_{q'}} \left( \left( \frac{P(T=1|X)P(X)}{Q(T=1|X)P(X \in B_{q'})} \right) \right)$$

$$= \sum_{X \in B_{q'}} \left( \left( \frac{P(T=1|X)P(X)}{q'P(X \in B_{q'})} \right) \right)$$

$$= \frac{P(T=1|X \in B_{q'}))}{q'}$$

Also, for the above data-generation process,

$$\tau = \hat{\tau}(P) = \sum_{X \in \mathcal{X}} (\mathbb{E}_{Y \sim P'(Y|X,do(T=1))}[Y] - \mathbb{E}_{Y \sim P'(Y|X,do(T=0))}[Y]).P(X)$$

$$= \sum_{q \in S_Q} \sum_{X \in B_q} (\mathbb{E}_{Y \sim P'(Y|X,do(T=1))}[Y] - \mathbb{E}_{Y \sim P'(Y|X,do(T=0))}[Y]).P(X)$$

$$= \sum_{X \in B_{q'}} P(X)/P(X \in B_{q'})$$

$$= 1$$

Thus,

$$\lim_{n \to \infty} \text{Probability}(\tau_n = \tau) = P(\hat{\tau}(Q) = \tau)$$

$$= \text{Probability} \left( \frac{P(T=1|X \in B_{q'})}{q'} = 1 \right)$$

$$= \text{Probability} \left( P(T=1|X \in B_{q'}) = q' \right)$$

$$= \text{Probability} \left( P(T=1|Q(T=1|X) = q') = q' \right)$$

$$= 0,$$

since we began with the assumption that $P(T=1|Q(T=1|X) = q') \neq q'$.

Please note that we could have defined a set of outcome functions that produce $Y = 0$ for $X \in B_{q'}$, thus, potentially letting us compute unbiased treatment effects despite working with a miscalibrated model. However, we want our IPTW estimator to provide unbiased ATE estimates over all possible outcome functions. Here, we can see that IPTW estimator for ATE that uses a miscalibrated propensity score model cannot obtain unbiased treatment effect estimates on all possible outcome functions.

$\square$

## H.3 CALIBRATED UNCERTAINTIES IMPROVE PROPENSITY SCORING MODELS

We define the true ATE as

$$\tau = \mathbb{E}_{y \sim P(Y=y|do(T=1))}[y] - \mathbb{E}_{y \sim P(Y=y|do(T=0))}[y]$$

$$= \sum_y y(\sum_X P(Y=y|X,do(T=1))P(X) - \sum_X P(Y=y|X,do(T=0))P(X))$$

$$= \sum_y y(\sum_X P(Y=y|X,T=1)P(X) - \sum_X P(Y=y|X,T=0)P(X))$$

Next, recall that the finite-sample Inverse Propensity of Treatment Weight (IPTW) estimator with a model $Q(T = 1|X)$ of $P(T = 1|X)$ produces an estimate $\hat{\tau}_n(Q)$ of the ATE, which is computed as

$$\hat{\tau}_n(Q) = \frac{1}{n}\sum_{i=1}^{n}\left(\frac{t^{(i)}y^{(i)}}{Q(T = 1|x^{(i)})} - \frac{(1 - t^{(i)})y^{(i)}}{1 - Q(T = 1|x^{(i)})}\right).$$

We define $\hat{\tau}(Q)$ as the limit $lim_{n\to\infty}\hat{\tau}_n(Q)$ when the amount of data goes to infinity. Notice that we can write

$$\lim_{n\to\infty}(\hat{\tau}_n(Q)) = \hat{\tau}(Q) = \sum_y y[\pi_{y,1}(Q) - \pi_{y,0}(Q)]$$

where

$$\pi_{y,t}(Q) = P(T = t)\sum_X P(Y = y|X, T = t)\frac{P(X|T = t)}{Q(T = t|X)} = \sum_X P(Y = y|X, T = t)\frac{P(T = t|X)}{Q(T = t|X)}P(X)$$

We have a multiplicative term $P(T = t)$ in the above expression since we are dividing by $n$ in the finite-sample formula as opposed to $n_t$ (the number of samples with treatment $t$). In other words, in order for the finite-sample formula to be a valid Monte Carlo estimator with samples coming from $P(X|T = t)$, there needs to be an "effective adjustment factor" of $n_t/n$ (such that $(n_t/n) \cdot (1/n_t) = (1/n)$), and this term is $P(T = t)$ in the limit of infinite data.

Clearly, if $Q = P$ we have $\hat{\tau}(Q) = \hat{\tau}(P) = \tau$. If not, we can consider the error

$$E = |(\hat{\tau}(P) - \hat{\tau}(Q))|.$$

### H.3.1    Bounding the Error of Causal Effect Estimation Using Proper Losses

We can form a bound on $E$ as

$$E = |[\hat{\tau}(P) - \hat{\tau}(Q)]|$$

$$= \left|\sum_y y[(\pi_{y,1}(P) - \pi_{y,0}(P)) - (\pi_{y,1}(Q) - \pi_{y,0}(Q))]\right|$$

$$\leq \sum_t \left|\sum_y y[(\pi_{y,t}(P) - \pi_{y,t}(Q)]\right|$$

$$\leq \sum_t \sum_y [|y||\pi_{y,t}(P) - \pi_{y,t}(Q)|]$$

$$= \sum_t \sum_y |y|[\left|\sum_X P(Y = y|X, T = t)P(X)\left(1 - \frac{P(T = t|X)}{Q(T = t|X)}\right)\right|]$$

$$\leq \sum_t \sum_y |y|[\sum_X P(Y = y|X, T = t)P(X)\left|1 - \frac{P(T = t|X)}{Q(T = t|X)}\right|]$$

$$= \sum_t \sum_y |y|.[\sum_X P(Y = y|X, T = t)P(X)\ell_X(P_t, Q_t)^{1/2}] \qquad \text{where } \ell_X(P_t, Q_t) = \left(1 - \frac{P(T = t|X)}{Q(T = t|X)}\right)^2$$

$$= \sum_t \sum_y |y|.\mathbb{E}_{X\sim R_{y,t}}[\ell_X(P_t, Q_t)^{1/2}]$$

where $R_{t,y} \propto P(Y = y|X, T = t)P(X)$ (i.e. $R_{t,y} \sim k.P(Y = y|X, T = t)P(X)$, $k$ is constant) and $\ell_X(P, Q)$ is a type of expected Chi-Squared divergence between $P, Q$. It is a type of proper score. Thus when $P = Q$, we get zero error, and otherwise we get a bound.

In the above derivation, we see that the expected error $|\pi_{y,t}(P) - \pi_{y,t}(Q)|$ induced by an IPTW estimator with propensity score model $Q$ is bounded as

$$|\pi_{y,t}(P) - \pi_{y,t}(Q)| \leq \mathbb{E}_{X\sim R_{y,t}}[\ell_\chi(P_t, Q_t)^{\frac{1}{2}}].$$

### H.3.2 Calibration Reduces Variance of Inverse Probability Estimators

**Theorem H.2.** *Let $P$ be the data distribution, and suppose that $1 - \delta > P(T|X) > \delta$ for all $T, X$ and let $Q$ be a calibrated model relative to $P$. Then $1 - \delta > Q(T|X) > \delta$ for all $T, X$ as well.*

*Proof.* Suppose $Q(T = 1|x) = q$ for some $x$ and $q < \delta$. Since $Q$ is calibrated, we have $P(T = 1|Q(T = 1|X) = q) = q < \delta$.

However $P(T = 1|x) > \delta$ for every $x$. Hence, $P(T = 1|X \in A) > \delta$, for all sets $A \subseteq \mathcal{X}$. This implies that $P(T = 1|Q(T = 1|X) = q) > \delta$ for all $q \in [0, 1]$.

Thus, we have a contradiction. $\qquad\square$

### H.3.3 Calibration Improves Error Bounds on Causal Effect Estimate

We show that calibration strictly improves our $\ell_\chi$ bound on the IPTW error.

**Theorem H.3.** *Let $\ell_1$ be the expected bound on the error of an uncalibrated IPTW estimator $Q_1$ in Corollary 3.4, and let $\ell_2$ be the bound for $Q_2$, the recalibrated version of $Q_1$ with $\ell_\chi^{1/2}$ as the choice of loss $L$ to train the recalibrator. Then as the size of the calibration set $n \to \infty$ we have $\ell_2 \leq \ell_1$ with equality iff $Q_1 = Q_2$.*

*Proof.* Corollary 3.4 states that the error of an IPTW estimator with propensity score model $Q$ is bounded by $2|\mathcal{Y}|K \max_{y,t} \mathbb{E}_{R_{y,t}} \ell_\chi(P, Q)^{\frac{1}{2}}$, where $|y| \leq K$ for all $y \in \mathcal{Y}$, $R_{y,t} \propto P(Y = y|X, T = t)P(X)$ is a data distribution and $\ell_\chi(P, Q) = \left(1 - \frac{P(T=t|X)}{Q(T=t|X)}\right)^2$ is the *chi*-squared loss between the true propensity score and the model $Q$.

Thus, $\ell_1 = 2|\mathcal{Y}|K \max_{y,t} \mathbb{E}_{R_{y,t}} \ell_\chi(P, Q_1)^{\frac{1}{2}}$ and $\ell_2 = 2|\mathcal{Y}|K \max_{y,t} \mathbb{E}_{R_{y,t}} \ell_\chi(P, Q_2)^{\frac{1}{2}}$. Clearly, the upper bound $\ell_i$ depends on $\ell_\chi(P, Q_i)$ where $i \in \{1, 2\}$.

When we use Algorithm 2 to perform recalibration, we obtain $Q_2 = R \circ Q_1$. Here, we can choose the loss function $L(Q, T) = \mathbb{E}_X \mathbb{E}_{T|X} \ell_\chi(Q(T = 1|X), T)^{1/2}$. From Theorem 4.4, it follows that $L(Q_2, T) = L(R \circ Q_1, T) \leq L(Q_1, T) + o(n)$ for a recalibrator $R$.

As $n \to \infty$, $R \to B$ (Bayes optimal recalibrator; see Task 4.1).

If $Q_1 \neq Q_2$, then $L(Q_2, T) \neq L(Q_1, T)$ because $L$ is strictly proper. Conversely, when $Q_1 = Q_2$ clearly $\ell_1 = \ell_2$. Hence, the claim follows. $\qquad\square$

Theorem H.6 in Appendix H.4 proves a similar result for the AIPW estimator when the outcome model is inaccurate.

### H.3.4 Calibration Improves the Accuracy of Causal Effect Estimation

Unfortunately, calibration by itself is not sufficient to correctly estimate treatment effects. For example, consider defining $Q(T|X)$ as the marginal $P(T)$: this $Q$ is calibrated, but cannot accurately estimate treatment effects. However, if the model $Q$ is sufficiently accurate (as might be the case with a powerful neural network), calibration becomes the missing piece for an accurate IPTW estimator.

Specifically, we define separability, a condition which states that when $P(T|X_1) \neq P(T|X_2)$ for $X_1, X_2 \in \mathcal{X}$, then the model $Q$ satisfies $Q(T|X_1) \neq Q(T|X_2)$. Intuitively, the model $Q$ is able to discriminate between various $T$—something that might be achievable with an expressive neural $Q$ that has high classification accuracy. We show that a model that is separable and also calibrated achieves accurate causal effect estimation.

**Theorem H.4.** *The error of an IPTW estimator with propensity model $Q$ tends to zero as $n \to \infty$ if:*

1. *Separability holds, i.e., $\forall X_1, X_2 \in \mathcal{X}, P(T|X_1) \neq P(T|X_2) \implies Q(T|X_1) \neq Q(T|X_2)$*

2. *The model $Q$ is calibrated, i.e., $\forall q \in (0, 1), P(T = 1|Q(T = 1|X) = q) = q$*

*Proof.* We prove this for discrete inputs at first and then prove it for continuous inputs.

**Discrete Input Space.**

If our input space $\mathcal{X}$ is discrete, then the number of distinct values that $Q(T = 1|X)$ can take is countable. Let us assume that $Q(T = 1|X)$ takes values $\{q_i\}_{i=1}^{M}$. Thus, we can partition $\mathcal{X}$ into buckets $\{B_i\}_{i=1}^{M}$ such that $B_i = \{X|Q(T = 1|X) = q_i\}$. Due to separability, we have $\forall X_1, X_2 \in \mathcal{X}, Q(T|X_1) = Q(T|X_2) \implies P(T|X_1) = P(T|X_2)$. Thus, we have $\forall i, \forall X_1, X_2 \in B_i, Q(T = 1|X_1) = Q(T = 1|X_2)$, and $P(T = 1|X_1) = P(T = 1|X_2)$.

Let us assume that for each bucket $B_i$, our true propensity $P(T = 1|X)$ is $p_i$, i.e, if $X \in B_i$ then $Q(T = 1|X) = q_i$ and $P(T = 1|X) = p_i$.

Assuming positivity, $0 < p_i < 1$.

Now, for all $i$, we can write

$$P(T = 1|Q(T = 1|X) = q_i) = P(T = 1|X \in B_i)$$
$$= p_i.$$

If $Q$ is calibrated, then by definition $p_i = q_i$.

Now, we can write the expression for ATE $\tau$ as

$$\tau = \hat{\tau}(P) = \mathbb{E}_{Y,T,X}\Big[\frac{TY}{P(T = 1|X)} - \frac{(1 - T)Y}{(1 - P(T = 1|X))}\Big]$$
$$= \sum_{i=1}^{N} P(X \in B_i)\mathbb{E}_{Y,T}\left(\frac{TY}{p_i} - \frac{(1 - T)Y}{(1 - p_i)}\right)$$

Using our propensity score model $Q(T = 1|X)$, we estimate $\hat{\tau}$ as

$$\hat{\tau}(Q) = \mathbb{E}_{Y,T,X}\Big[\frac{TY}{Q(T = 1|X)} - \frac{(1 - T)Y}{(1 - Q(T = 1|X))}\Big]$$
$$= \sum_{i=1}^{N} P(X \in B_i)\mathbb{E}_{Y,T}\left(\frac{TY}{q_i} - \frac{(1 - T)Y}{(1 - q_i)}\right)$$

If our model $Q$ is calibrated, then $p_i = q_i$. Hence, $0 < q_i < 1$ and $\hat{\tau}$ is well-defined. Also, $\tau = \hat{\tau}(P) = \hat{\tau}(Q)$.

When our observational data contains $n$ units, the IPTW estimator based on model $Q(T = 1|X)$ is $\hat{\tau}_n = \frac{1}{n}\sum_{i=0}^{n}\left(\frac{T^{(i)}Y^{(i)}}{Q(T=1|X^{(i)})} - \frac{(1-T^{(i)})Y^{(i)}}{1-Q(T=1|X^{(i)})}\right)$.

Hence, $\lim_{n\to\infty} \hat{\tau}_n = \hat{\tau}(Q) = \hat{\tau}(P) = \tau$.

**Continuous Input Space.**

When $X$ is continuous, the number of buckets can be uncountable. The buckets can now be formed as $B_q = \{X|Q(T = 1|X) = q\}, \forall q \in [0, 1]$. It is easy to see that $\{B_q\}_{q\in[0,1]}$ partitions $\mathcal{X}$. Note that $B_q$ can be empty if there exists no $X$ such that $Q(T = 1|X) = q$.

Due to separability, $\forall X_1, X_2 \in \mathcal{X}, Q(T|X_1) = Q(T|X_2) \implies P(T|X_1) = P(T|X_2)$. Thus, for all $q$, $P(T = 1|X)$ takes on a unique value for all $X \in B_q$, i.e., $\forall q \in [0, 1], P(T = 1|X \in B_q) = f(q)$, where function $f : [0, 1] \to [0, 1]$.

Hence, we can write

$$\forall q \in [0, 1], P(T = 1|Q(T = 1|X) = q) = P(T = 1|X \in B_q)$$
$$= f(q).$$

When model $Q(T = 1|X)$ is calibrated by our definition, then $\forall q \in [0, 1], q = f(q)$.

Therefore, $\forall q \in [0,1], Q(T=1|X \in B_q) = q = f(q) = P(T=1|X \in B_q)$.

Since $\{B_q\}_{q \in [0,1]}$ partitions $\mathcal{X}$, we have $\forall X \in \mathcal{X}, P(T=1|X) = Q(T=1|X)$. Thus, $\hat{\tau}(P) = \hat{\tau}(Q)$.

$\square$

## H.4 DOUBLY ROBUST ESTIMATORS AND ERROR BOUNDS ON CAUSAL EFFECT ESTIMATION

Given a dataset $\{x_i, t_i, y_i\}_{i=1}^n$, the doubly robust AIPW (Augmented Inverse Propensity Weight) estimator can be used to compute ATE estimate as

$$\hat{\tau}'_n(Q, f) = \frac{1}{n} \sum_{i=1}^n \left( f(x_i, 1) - f(x_i, 0) + \frac{t^{(i)}(y^{(i)} - f(x_i, 1))}{Q(T=1|x^{(i)})} - \frac{(1-t^{(i)})(y^{(i)} - f(x, 0))}{1 - Q(T=1|x^{(i)})} \right).$$

The outcome model $f(X=x, T=t)$ can be learned from available data to predict potential outcome $Y[X=x, do(T=t)]$, where the input covariates are set to $x$ and the applied intervention is $T=t$. Let us assume that $f(X=x, T=t)$ produces an error of $\epsilon(X=x, T=t)$, i.e. $f(X, T) = Y[X, do(T)] + \epsilon(X, T)$.

Thus, we can rewrite the causal effect estimate $\hat{\tau}'_n(Q, \epsilon)$ as

$$\hat{\tau}'_n(Q, \epsilon) = \frac{1}{n} \sum_{i=1}^n \left( Y[x_i, do(t=1)] - Y[x_i, do(t=0)] + \epsilon(x_i, 1) - \epsilon(x_i, 0) - \frac{t^{(i)}(\epsilon(x_i, 1))}{Q(T=1|x^{(i)})} + \frac{(1-t^{(i)})(\epsilon(x, 0))}{1 - Q(T=1|x^{(i)})} \right).$$

When $n \to \infty$, we have

$$\lim_{n \to \infty} \hat{\tau}'_n(Q, \epsilon) = \hat{\tau}'(Q, \epsilon) = \hat{\tau}'(Q, 0) + \mathbb{E}_{X,T}[\epsilon(X, 1)(1 - \frac{T}{Q(T=1|X)} - \epsilon(X, 0)(1 - \frac{1-T}{1-Q(T=1|X)}))],$$

where second equality is true due to doubly robust property. We state the following error bound for the AIPW estimator:

**Corollary H.5.** *Let $|\epsilon(X, T)| \le \epsilon_{max}$ for all $X \in \mathcal{X}, T \in \{0, 1\}$. The error of an AIPW estimator with propensity score model $Q$ and error in outcome model $\epsilon$ is bounded by $\epsilon_{max} \sum_t \mathbb{E}_X[l_X(P_t, Q_t)^{1/2}]$ where $P_t = P(T=t|X), Q_t = Q(T=t|X)$.*

Due to the doubly robust property, we know that the true ATE estimate $\tau = \hat{\tau}'(Q, 0) = \hat{\tau}'(P, \epsilon)$ for any propensity model $Q(T=1|X)$ and error function $\epsilon(X, T)$.

The L1 error in our ATE estimate $\hat{\tau}'_n(Q, \epsilon)$ (after seeing infinite samples) can be expressed as

$$E = |\hat{\tau}'(Q, \epsilon) - \tau| = |\hat{\tau}'(Q, \epsilon) - \hat{\tau}'(Q, 0)|$$

Thus,

$$E = \left| \mathbb{E}_{X,T}[\epsilon(X, 1)(1 - \frac{T}{Q(T=1|X)} - \epsilon(X, 0)(1 - \frac{1-T}{1-Q(T=1|X)}))] \right|$$

$$= \left| \mathbb{E}_X \mathbb{E}_{T|X}[\epsilon(X, 1)(1 - \frac{T}{Q(T=1|X)} - \epsilon(X, 0)(1 - \frac{1-T}{1-Q(T=1|X)}))] \right|$$

$$= \left| \mathbb{E}_X[\epsilon(X, 1)(1 - \frac{P(T=1|X)}{Q(T=1|X)} - \epsilon(X, 0)(1 - \frac{1-P(T=1|X)}{1-Q(T=1|X)}))] \right|$$

$$\le \mathbb{E}_X \left[ \left| \epsilon(X, 1)(1 - \frac{P(T=1|X)}{Q(T=1|X)} \right| + \left| \epsilon(X, 0)(1 - \frac{P(T=0|X)}{Q(T=0|X)})) \right| \right]$$

$$\le \epsilon_{max} \mathbb{E}_X \left[ \left| (1 - \frac{P(T=1|X)}{Q(T=1|X)} \right| + \left| (1 - \frac{P(T=0|X)}{Q(T=0|X)})) \right| \right] \qquad \text{where } \epsilon_{max} = \max_{X,T} |\epsilon(X, T)|$$

$$\le \epsilon_{max} \sum_t \mathbb{E}_X[l_X(P_t, Q_t)^{1/2}] \qquad \text{where } P_t = P(T=t|X), Q_t = Q(T=t|X)$$

Thus, we have an error bound on the asymptotic ATE estimate that relates with the chi-squared divergence. Thus, given that the learned outcome model is inaccurate (due to possible mis-specification), training a recalibrator for the propensity score model with $l_X$ as loss function reduces the chi-squared divergence and improves the error bound.

**Theorem H.6.** *Let $\ell$ be the expected bound on the error of an uncalibrated AIPW estimator $Q$ in Corollary H.5, and let $\ell'$ be the bound for $Q'$, the recalibrated version of $Q$ obtained using Algorithm 2 with $\ell_\chi^{1/2}$ as the choice of loss L. Then as the size of the calibration set $n \to \infty$ we have $\ell' \leq \ell$ with equality iff $Q = Q'$.*

*Proof.* Corollary H.5 states that the error of an AIPW estimator with propensity score model $Q$ and error in outcome model $\epsilon$ is bounded by $\epsilon_{max} \sum_t \mathbb{E}_X[l_X(P_t, Q_t)^{1/2}]$ where $|\epsilon(X,T)| \leq \epsilon_{max}$ for all $X \in \mathcal{X}, T \in \{0,1\}$, $P_t = P(T=t|X), Q_t = Q(T=t|X)$ and $\ell_\chi(P_t, Q_t) = \left(1 - \frac{P(T=t|X)}{Q(T=t|X)}\right)^2$ is the *chi*-squared loss between the true propensity score and the model $Q$.

Thus, $\ell = \epsilon_{max} \sum_t \mathbb{E}_X[l_X(P_t, Q_t)^{1/2}]$ and $\ell' = \epsilon_{max} \sum_t \mathbb{E}_X[l_X(P_t, Q'_t)^{1/2}]$. Clearly, the upper bound on $\ell$ and $\ell'$ depends on $\ell_\chi(P, Q)$ and $\ell_\chi(P, Q')$ respectively.

When we use Algorithm 2 to perform recalibration, we obtain $Q' = R \circ Q$. Here, we can choose the loss function $L(Q, T) = \mathbb{E}_X \mathbb{E}_{T|X} \ell_\chi(Q(T=1|X), T)^{1/2}$. From Theorem 4.4, it follows that $L(Q', T) = L(R \circ Q, T) \leq L(Q, T) + o(n)$ for a recalibrator $R$.

As $n \to \infty$, $R \to B$ (Bayes optimal recalibrator; see Task 4.1).

If $Q \neq Q'$, then $L(Q', T) \neq L(Q, T)$ because $L$ is strictly proper. Conversely, when $Q = Q'$ clearly $\ell = \ell'$. Hence, the claim follows. $\square$

Now, we prove that calibration is a necessary condition for accurate causal effect estimation when the outcome model in AIPW estimator is inaccurate.

**Theorem H.7.** *When propensity model $Q(T|X)$ is not calibrated and the outcome model f(X, T) is inaccurate for $X \in \{X : Q(T=1|X) = q\} \subseteq \mathcal{X}$ such that $q \in (0,1), P(T=1|Q(T=1|X') = q) \neq q$, then there exists true outcome function such that the doubly robust AIPW estimator based on Q and f yields an incorrect estimate of true causal effects almost surely.*

*Proof.* Following the setup in H.2, we let $S_Q = \{q | \exists X \in \mathcal{X}, Q(T=1|X) = q\}$. We partition $\mathcal{X}$ into buckets $\{B_q\}_{q \in S_q}$ such that $B_q = \{X | Q(T=1|X) = q\}$. When $Q(T=1|X)$ is not calibrated, we know that $\exists q \in [0,1], P(T=1|Q(T=1|X) = q) \neq q$.

We design the true outcome function $Y[X, do(T=t)]$ such that $Y[X, do(T=0)] = 0$. Since the learned outcome model $f(X, T) = Y[X, do(T=t)] + \epsilon(X, T)$ is inaccurate (possibly from learning a mis-specified model), let us define $\mathcal{X}_\epsilon \subseteq \mathcal{X}$ such that $\forall X \in \mathcal{X}_\epsilon, \epsilon(X,T) \neq 0$ and $\forall X \in \mathcal{X}/\mathcal{X}_\epsilon, \epsilon(X,T) = 0$. For the sake of simplicity, we assume that the outcome model $f(x,t)$ can learn the true outcome function whenever $T = 0$, since the true outcome is 0 whenever $T = 0$, Thus, $\forall X \in \mathcal{X}, \epsilon(X, T = 0) = 0$.

Now, let us pick $q' \in S_Q$ such that $P(T=1|Q(T=1|X) = q') \neq q'$ and $B_{q'} \cap \mathcal{X}_\epsilon \neq \phi$. We can always pick such a $q'$ as long as $Q$ is uncalibrated and $\exists X \in B_{q'}$ such that , $\epsilon(X, T=1) \neq 0$ (i.e. the learned outcome model $f(X,T)$ is inaccurate where the learned propensity model produces inaccurate uncertainties).

With this, we can write the expression for PEHE (Precision in Estimation of Heterogenous Treatment Effect) estimate with $n$ samples $F_n(Q, \epsilon)$ as

$$F_n(Q, \epsilon) = \frac{1}{n}\sum_{i=1}^n \left( Y[x_i, do(t=1)] - Y[x_i, do(t=0)] + \epsilon(x_i, 1) - \epsilon(x_i, 0) - \frac{t^{(i)}(\epsilon(x_i, 1))}{Q(T=1|x^{(i)})} + \frac{(1-t^{(i)})(\epsilon(x,0))}{1-Q(T=1|x^{(i)})} - \right.$$

$$\left. (Y[x_i, do(t=1)] - Y[x_i, do(t=0)]) \right)^2 = \frac{1}{n}\sum_{i=1}^n \left( \epsilon(x_i, 1) - \epsilon(x_i, 0) - \frac{t^{(i)}(\epsilon(x_i, 1))}{Q(T=1|x^{(i)})} + \frac{(1-t^{(i)})(\epsilon(x,0))}{1-Q(T=1|x^{(i)})} \right)^2.$$

Now, we will try to establish a lower bound on the error $F_n(Q, \epsilon)$ when $n \to \infty$.

$$F = \lim_{n \to \infty} F_n(Q, \epsilon)$$

$$= \mathbb{E}_{X,T}[(\epsilon(X,1)(1 - \frac{T}{Q(T=1|X)} - \epsilon(X,0)(1 - \frac{1-T}{1-Q(T=1|X)})))^2]$$

$$= \mathbb{E}_X \mathbb{E}_{T|X}[(\epsilon(X,1)(1 - \frac{T}{Q(T=1|X)} - \epsilon(X,0)(1 - \frac{1-T}{1-Q(T=1|X)})))^2]$$

Following the setup in H.2, we expand the expectation over X

(similar expression can be written with $\int_X$ if X is continuous)

$$= \sum_{q \in S_Q} \sum_{X \in B_q} (\epsilon(X,1)(1 - \frac{P(T=1|X)}{Q(T=1|X)} - \epsilon(X,0)(1 - \frac{1-P(T=1|X)}{1-Q(T=1|X)})))^2 P(X)$$

$$= \sum_{q \in S_Q} \sum_{X \in B_q \cap \mathcal{X}_\epsilon} (\epsilon(X,1)(1 - \frac{P(T=1|X)}{Q(T=1|X)} - \epsilon(X,0)(1 - \frac{1-P(T=1|X)}{1-Q(T=1|X)})))^2 P(X) \qquad \forall X \in \mathcal{X}/\mathcal{X}_\epsilon, \epsilon(X,T) = 0$$

Since we assume that $\forall x \in \mathcal{X}, \epsilon(x,0) = 0$,

$$= \sum_{q \in S_Q} \sum_{X \in B_q \cap \mathcal{X}_\epsilon} (\epsilon(X,1)(1 - \frac{P(T=1|X)}{q})^2 P(X)$$

$$\geq \sum_{X \in B_{q'} \cap \mathcal{X}_\epsilon} (\epsilon(X,1)(1 - \frac{P(T=1|X)}{q'})^2 P(X) \qquad P(T=1|Q(T=1|X) = q') \neq q'$$

$$\geq \epsilon_{min} \sum_{X \in B_{q'} \cap \mathcal{X}_\epsilon} ((1 - \frac{P(T=1|X)}{q'})^2 P(X) \qquad \epsilon_{min} = \min_{X \in B_q \cap \mathcal{X}_\epsilon} \epsilon(X,1)$$

The above expression is non-zero since $P(T=1|Q(T=1|X) = q') \neq q'$ and $\epsilon_{min} \neq 0$ by design. Thus, when $Q(T|X)$ is not calibrated and the learned outcome model f(X, T) is inaccurate over the regions where $P(T=1|Q(T=1|X) = q) \neq q$, then there exists true outcome function such that the AIPW estimator based on Q and f yields an incorrect estimate of true causal effects almost surely. □

## H.5 ALGORITHMS FOR CALIBRATED PROPENSITY SCORING

### H.5.1 Asymptotic Calibration Guarantee

**Theorem H.8.** *The model $R \circ Q$ is asymptotically calibrated and the calibration error $\mathbb{E}[L_c(R \circ Q, S)] < \delta(m)$ for $\delta(m) = o(m^{-k}), k > 0$ w.h.p.*

*Proof.* Any proper loss can be decomposed as: proper loss = calibration - sharpness + irreducible term [Guo et al., 2017]. The calibration term consists of the error $\mathbb{E}[L_c(R \circ Q, S)]$. The sharpness and irreducible term can be represented as the refinement term $\mathbb{E}(L_r(S))$. Table 10 provides examples of some proper loss functions and the respective decompositions. The rest of our proof uses the techniques of Kuleshov and Deshpande [2022] in the context of propensity scores.

Kull and Flach [2015] show that the refinement term can be further divided as $\mathbb{E}(L_r(S)) = \mathbb{E}(L_g(S, B \circ Q)) + \mathbb{E}(L(B \circ Q, T))$. Here, $B$ is the Bayes optimal recalibrator $P(T=1|Q(T=1|X))$ and $S$ is $P(T=1|R \circ Q)$.

| Proper Score | Loss $L(F,G)$ | Calibration $L_c(F,S)$ | Refinement $L_r(S)$ |
|---|---|---|---|
| Logarithmic | $\mathbb{E}_{y\sim G}\log f(y)$ | $KL(s\|f)$ | $H(s)$ |
| CRPS | $\mathbb{E}_{y\sim G}(F(y)-G(y))^2$ | $\int_{-\infty}^{\infty}(F(y)-S(y))^2 dy$ | $\int_{-\infty}^{\infty}S(y)(1-S(y))dy$ |
| Quantile | $\mathbb{E}_{y\sim G}^{\tau\in U[0,1]}\rho_\tau(y-F^{-1}(\tau))$ | $\int_0^1\int_{S^{-1}(\tau)}^{F^{-1}(\tau)}(S(y)-\tau)dyd\tau$ | $\mathbb{E}_{y\sim S}^{\tau\in U[0,1]}\rho_\tau(y-S^{-1}(\tau))$ |

Table 10: Proper loss functions. A proper loss is a function $L(F,G)$ over a forecast $F$ targeting a variable $y\in\mathcal{Y}$ whose true distribution is $G$ and for which $S(F,G)\geq S(G,G)$ for all $F$. Each $L(F,G)$ decomposes into the sum of a calibration loss term $L_c(F,S)$ (also known as reliability) and a refinement loss term $L_r(S)$ (which itself decomposes into sharpness and an uncertainty term). Here, $S(y)$ denotes the cumulative distribution function of the conditional distribution $\mathbb{P}(Y=y\mid F_X=F)$ of $Y$ given a forecast $F$, and $s(y),f(y)$ are the probability density functions of $S$ and $F$, respectively. We give three examples of proper losses: the log-loss, the continuous ranked probability score (CRPS), and the quantile loss.

Recall that if we solve the Task 4.1, we have for $\delta(m)=o(1)$

$$\mathbb{E}(L(B\circ Q,T))\leq\mathbb{E}(L(R\circ Q,T))\leq\mathbb{E}(L(B\circ Q,T))+\delta(m)$$

Using Gneiting et al. [2007], Kull and Flach [2015] we decompose $\mathbb{E}(L(R\circ Q,T))$

$$\implies \mathbb{E}(L(B\circ Q,T))\leq\mathbb{E}(L_c(R\circ Q,S))+\mathbb{E}(L_g(S,B\circ Q))+\mathbb{E}(L(B\circ Q,T))\leq\mathbb{E}(L(B\circ Q,T))+\delta(m)$$
$$\implies \mathbb{E}(L_c(R\circ Q,S))+\mathbb{E}(L_g(S,B\circ Q))\leq\delta(m)$$
$$\implies \mathbb{E}(L_c(R\circ Q,S))\leq\delta(m)$$

Thus, solving Task 4.1 allows us to obtain asymptotically calibrated $R\circ Q$ such that the calibration error is bounded as $\mathbb{E}[L_c(R\circ Q,S)]<\delta(m)$.

$\square$

### H.5.2 No-Regret Calibration

**Theorem H.9.** *The recalibrated model has asymptotically vanishing regret relative to the base model:* $\mathbb{E}[L(R\circ Q,T)]\leq\mathbb{E}[L(Q,T)]+\delta$, *where* $\delta>0,\delta=o(m^{-k}),k>0$.

*Proof.* Solving Task 4.1 implies $\mathbb{E}[L(R\circ Q,T)]\leq\mathbb{E}[L(B\circ Q,T)]+\delta\leq\mathbb{E}[L(Q,T)]+\delta$. The first inequality comes from definition of Task 4.1 and the second inequality holds because a Bayes-optimal $B$ has lower loss than an identity mapping [Kuleshov and Deshpande, 2022].

$\square$