# OpenReview forum: "Calibrated and Conformal Propensity Scores for Causal Effect Estimation"
_auai.org/UAI/2024/Conference — UAI 2024 poster_

### Official Review · Reviewer_JPin · 2024-03-11

**Q2-1 Originality-Novelty:** 3
**Q2-2 Correctness-Technical Quality:** 3
**Q2-5 Clarity Of Writing:** 3

**Q1 Summary And Contributions:**

This work studies propensity scores of treatment effect estimation and proposed that calibration is needed for propensity score based CATE estimators. It uses proof by contradiction to show that calibration is a necessary condition for IPTW/AIPW estimators. After theoretical bound on the true and estimated casual effects has been given based on the difference between the true and estimated propensities, the authors showed that calibration can help reduce variances and improve error bounds. The authors then proposed an algorithm utilize a recalibrator based on a hold-out calibration set of samples. Evaluation conducted on a wide range of datasets demonstrated the effectiveness of the proposed propensity score calibration method.

**Q2-3 Extent To Which Claims Are Supported By Evidence:**

3: Good: the main claims are supported by convincing evidence (in the form of adequate experimental evaluation, proofs, (pseudo-)code, references, assumptions).

**Q2-4 Reproducibility:**

2: Fair: key resources (e.g. proofs, code, data) are unavailable but key details (e.g. proof sketches, experimental setup) are sufficiently well-described for an expert to confidently reproduce the main results.

**Q3 Main Strengths:**

1. This paper starts with a well-motivated problem of uncalibrated propensity score and provides both theoretical and practical approaches to calibration.
2. Experimental evaluations are well designed and covers a wide scenario of datasets.
2. The main text of the paper is well-written and easy to follow.

**Q4 Main Weakness:**

Using a separate hold-out set for calibration may reduce the size of samples under finite sample setting, which may reduce the effectiveness of propensity calibration. However, this is not sufficiently discussed in the paper and no theoretical guidelines have been provided to balance the trade-off.

**Q5 Detailed Comments To The Authors:**

Some typesettings in the bibliography are incorrect and should be rectified and carefully proofread before publication.

**Q9 Complying With Reviewing Instructions:**

Yes

---

> ### Author Rebuttal · Authors · 2024-04-08
>
> We thank the reviewer for their positive comments and constructive feedback. Below we respond to the concerns and questions raised.
>
> __Need for Separate Hold-out Dataset__
>
> The requirement to allocate a separate calibration dataset may indeed reduce the size of dataset available for training the propensity score model Q(T|X). However, we can use cross-validation splits in the dataset to calibrate a propensity score model, as mentioned in Section 5 under Setup.
>
> To implement this approach, we divide our dataset D into $k$ partitions $\{S_1, S_2,..,S_k\}$. For each dataset split $S_k$, we train the propensity score model $Q_k(T|X)$ on $S_k$ and and generate a partial recalibrator training dataset (as defined in Algorithm 2) as $C_k = \\{Q_k(x), y | x, y \in D - S_k\\}$. After this, we can take a union over all $C_k$ to generate the final recalibrator training dataset. This allows us to use the entire available dataset for training the propensity score model as well as the recalibrator. This can be useful especially when the available dataset size is small. In our experiments, we have used leave-one-out cross-validation splits (thus, each partition $S_k$ is of size n-1 where n is the size of dataset D). We will add a discussion on this in the main paper.
>
>
> We thank the reviewer for pointing out typesetting issues in the bibliography. We will address this in the camera-ready version of paper.

---

### Official Review · Reviewer_KJk5 · 2024-03-19

**Q2-1 Originality-Novelty:** 2
**Q2-2 Correctness-Technical Quality:** 2
**Q2-5 Clarity Of Writing:** 2

**Q1 Summary And Contributions:**

This paper provides the necessity of using the calibrated propensity score, provide a method for estimating the calibrated propensity score, and offers the learning guarantees.

**Q2-3 Extent To Which Claims Are Supported By Evidence:**

2: Fair: the main claims are somewhat supported by evidence (but the experimental evaluation may be weak, or does not match entirely with the claims, important baselines may be missing, proofs contain important ideas but lack rigor, algorithmic details are only discussed superficially, references are imprecise, assumptions are not sufficiently motivated or explicated, etc.).

**Q2-4 Reproducibility:**

3: Good: key resources (e.g. proofs, code, data) are available and key details (e.g. proofs, experimental setup) are sufficiently well-described for competent researchers to confidently reproduce the main results.

**Q3 Main Strengths:**

1. Good demonstration when explaining technical concepts.

**Q4 Main Weakness:**

1. Weak literature review
2. There are some redundant parts unrelated with the main result.
3. Unclear distinction between calibrated propensity score versus well-specified (accurate) propensity score.
4. Some disconnection in the writing
5. Weak significance — Main results look pretty obvious

**Q5 Detailed Comments To The Authors:**

**Weak literature review**

1. Literature review is weak. I want to know more about calibration, but it’s hard to track how the calibration theories have been evolved.
2. I think “Causal Isotonic Calibration for Heterogeneous Treatment Effects” is also an important paper in the intersection between causal inference and calibration, but this paper is not citied.

**Redundancy**

1. The conformal prediction in Section 2.2. looks sudden and unrelated with the main result.
2. In Section 4.1., there is a function H. Is this function defined?

**Unclear distinction of notions & Weak significance**

1. I think the uncalibrated propensity score and the inaccurate propensity score are interchangeably used. Specifically, in Example in Theorem 3.1, it seems to me that Q(T | X) is not only uncalibrated but also ill-specified (inaccurate). For the clear distinction, I think it’s desirable to use the example Q(T | X) as a consistent but uncalibrated model. In the current shape where the inaccurate propensity score is the same as the uncalibrated model, Theorem 3.1, Theorem 3.2 and Theorem 3.5 look quite obvious and insignificant.

**Q9 Complying With Reviewing Instructions:**

Yes

---

> ### Author Rebuttal · Authors · 2024-04-08
>
> We thank the reviewer for their positive comments and constructive feedback. Below we respond to the concerns and questions raised.
>
> __Concern 1: Literature Review on Calibration and Causal Inference__
>
> Please see our response to reviewer CYwf for comparison against ‘Causal Isotonic Calibration’ (Van Der Laan et. al 2023 [1]) and ‘Calibration error for heterogeneous treatment effects’ (Xu et al 2022[2])
>
> In short, Van Der Laan et. al [1] enforce a __different notion of calibration__ on the Heterogeneous Treatment Effect (HTE) prediction: The average HTE of units with a given predicted HTE is equal to the shared predicted value. Our work calibrates the uncertainty outcome of the propensity score model such that the number of units receiving treatment, given X % predicted probability of receiving treatment, is equal to X %. While causal isotonic regression[1] performs calibration of HTEs more directly, it requires the propensity score model or outcome model to be sufficiently accurate for their approach to work. Additionally, their experimental setup ensures that their propensity score models are consistent and do not produce extreme values. Ensuring this is difficult in practice, especially when we don't know the true treatment assignment mechanism. Uncertainty calibration can be applied to misspecified propensity models (possibly producing extreme weights). Although we only present results with the ATE metric in our main paper, our method applies to both ATE and HTE estimation. Table 6 in our Appendix demonstrates that propensity score calibration also improves HTE estimation consistently.
>
> We will cite and discuss this work in the paper.
>
> _Discussing literature on calibration_
>
> In the context of causal inference, different notions of calibration have been studied to improve treatment effect estimation, e.g., calibration for improving heterogeneous treatment effect estimation [1,2] (discussed above), covariate balancing calibration [3] (please see our response to reviewers v3iY for a discussion on covariate balancing), calibration to correct for unobserved confounding [4], etc. However, uncertainty calibration of the propensity score model and its role in accurate causal effect estimation hasn’t been analyzed thoroughly.
>
> In general, Platt scaling [Platt, 1999] and isotonic regression [Niculescu-Mizil and Caruana, 2005] have been used to calibrate the uncertainties over binary classification outcomes. Uncertainty calibration has been extended to multi-class classification [Zadrozny et al. 2002], regression [Kuleshov et al. 2018] and online learning [Kuleshov and Ermon, 2017]. We focus on the discrete treatment regime in our paper, but it is possible to extend our method to continuous treatments as future work. Calibration has been applied to improve the accuracy of uncertainties in reinforcement learning [Malik et al., 2019], natural language processing [Nguyen & O’Connor, 2015], machine translation [Kumar and Sarawagi, 2019], Bayesian optimization [Deshpande and Kuleshov, 2023], etc.
>
> We will discuss the relevant literature on calibration in more details in the final version of the paper.
>
> [1] Van Der Laan L, Ulloa-Pérez E, Carone M, et al. Causal isotonic calibration for heterogeneous treatment effects[C]//International Conference on Machine Learning. PMLR, 2023: 34831-34854.
>
> [2] Xu Y, Yadlowsky S. Calibration error for heterogeneous treatment effects[C]//International Conference on Artificial Intelligence and Statistics. PMLR, 2022: 9280-9303.
>
> [3] Z Tan. Regularized calibrated estimation of propensity scores with model misspecification and high-dimensional data.
>
> [4] Til Stürmer, Sebastian Schneeweiss, Kenneth J Rothman, Jerry Avorn, and Robert J Glynn. Performance of propen- sity score calibration–a simulation study. Am. J. Epi- demiol., 165(10):1110–1118, May 2007b.
>
>
> __Concern 2: Uncalibrated and Inaccurate Q(T|X) Used Interchangeably, Making Theoretical Results Less Interesting__
>
> While discussing Theorem 3.1 in the paper, we use a simple binary setup to show that when the model $Q(T|X)$ is uncalibrated, it is always possible to pick a true outcome function such that the predicted causal effect is inaccurate. In this simple binary example, the lack of calibration implies that the outcome of Q(T|X) would be inaccurate for X=0 or X=1. Our theoretical results indeed only require the model to be uncalibrated (and not inconsistent) as also reflected in the proofs (Appendix H). We will clarify this further in the final version of the paper.
>
>
> __Concern 3: Redundancy__
>
> Thank you for pointing out the typo in Section 4.1. Instead of H, we would like to refer to propensity model Q as the base model. We will correct this in the final version.
>
> Conformal prediction can be used to obtain similar calibration guarantees for a single confidence interval instead of the full uncertainty distribution. We will expand this section to discuss how conformal prediction relates with our work.

---

### Official Review · Reviewer_7D7j · 2024-03-22

**Q2-1 Originality-Novelty:** 3
**Q2-2 Correctness-Technical Quality:** 3
**Q2-5 Clarity Of Writing:** 3

**Q10 Ethical Concerns:**

No.

**Q1 Summary And Contributions:**

The paper studies the importance of having a well-calibrated propensity score model in the context of propensity-weighted causal effect estimation. Theoretically, in the considered context, the paper: (i) shows that calibration is a necessary condition for a correct propensity score model, (ii) derives error bounds of the causal effect estimate based on the quality of the uncertainties of the propensity score model and shows that calibration improves the bound, (iii) shows that calibration can reduce variance of estimators caused by extreme propensity weights. Then, the paper proposes an algorithm for learning calibrated propensity score models, with theoretical guarantees on calibration and regret. Finally, the utility of the algorithm is demonstrated in extensive numerical studies.

**Q2-3 Extent To Which Claims Are Supported By Evidence:**

4: Excellent: all claims are supported by very convincing evidence (in the form of comprehensive experimental evaluation, rigorous mathematical proofs, detailed (pseudo-)code, precise references, well-motivated and realistic assumptions) and the authors deliver what they promise.

**Q2-4 Reproducibility:**

3: Good: key resources (e.g. proofs, code, data) are available and key details (e.g. proofs, experimental setup) are sufficiently well-described for competent researchers to confidently reproduce the main results.

**Q3 Main Strengths:**

A very good paper that considers an important problem in causal effect estimation. It has interesting theoretical results related to the considered problem, a conceptually simple algorithm (with theoretical guarantees), and convincing numerical experiments.

**Q4 Main Weakness:**

The presentation could perhaps be improved at some places. In particular, some of the in-text mathematical formulas are hard to read, especially when broken up over two lines.

**Q5 Detailed Comments To The Authors:**

Page 2: the part about "Calibrated and Conformal Prediction" could be extended a bit.

Page 3 - Thm 3.2: "..exists a true.." (typo)

Page 5: "utilize cross-validation splits to generate the calibration dataset" <- Could you elaborate a bit about what this means?

Page 8: "of the underlying" (typo)

Page 8 - Table 5: Do you have any intuition of why NB, which performs very poorly compared to LR, outperforms LR after both methods are calibrated?

Sometimes active referencing is used when passive is intended.

**Q9 Complying With Reviewing Instructions:**

Yes

---

> ### Author Rebuttal · Authors · 2024-04-08
>
> We thank the reviewer for their positive comments and constructive feedback. Below we respond to the concerns and questions raised.
>
>
> __Concern 1: Cross-validation data splits for calibration__
>
> The requirement to allocate a separate calibration dataset may reduce the size of dataset available for training the propensity score model $Q(T|X)$. Hence, we can use cross-validation splits in the dataset to calibrate a propensity score model, as mentioned in Section 5 under Setup.
>
> To implement this approach, we divide our dataset D into $k$ partitions $\{S_1, S_2,..,S_k\}$. For each dataset split $S_k$, we train the propensity score model $Q_k(T|X)$ on $S_k$ and and generate a partial recalibrator training dataset (as defined in Algorithm 2) as $C_k = \\{Q_k(x), y | x, y \in D - S_k\\}$. After this, we can take a union over all $C_k$ to generate the final recalibrator training dataset. This allows us to use the entire available dataset for training the propensity score model as well as the recalibrator. This can be useful especially when the available dataset size is small. In our experiments, we have used leave-one-out cross-validation splits (thus, each partition $S_k$ is of size n-1 where n is the size of dataset D). We will add a discussion on this in the main paper.
>
>
> __Concern 2: Additional Questions and Comments__
>
> _Extending calibrated and conformal prediction_
>
> We will extend this section to include more details on the definition of calibration (including modifications in case of multiple/continuous treatments). Conformal prediction can be used to obtain similar calibration guarantees for a single confidence interval instead of the full uncertainty distribution. We will expand this section to discuss how conformal prediction relates with our work.
>
> _Table 5: Why does NB, which performs very poorly compared to LR, outperforms LR after both methods are calibrated?_
>
> In general, we observe that calibrated NB and calibrated LR produce comparable ATE errors, as shown in Table 9 in Appendix G (which compares the performance of calibration across multiple base propensity models and various GWAS datasets).  In Table 5, when we consider the error bars, ATE error produced by calibrated NB is still comparable to the one produced by calibrated LR. Plain Naive Bayes performs very poorly as compared to Logistic Regression as it is a very simple model and the independence assumption within the model does not hold in our experimental setup. In Table 5, we show that calibration improves the performance of both LR and NB models (comparable to the standard LMM method for GWAS). Since NB is much faster to train as compared to LR for high-dimensional covariates, we show that calibration enables the use of a simple model like NB for a computationally intensive GWAS task.
>
> _Suggestions on presentation and typos_
>
> We thank the reviewer for pointing out typos. We will fix these in addition to improving the readability of in-line Math formulas.

---

### Official Review · Reviewer_CYwF · 2024-03-23

**Q2-1 Originality-Novelty:** 3
**Q2-2 Correctness-Technical Quality:** 3
**Q2-5 Clarity Of Writing:** 3

**Q1 Summary And Contributions:**

-	This paper investigates a critical issue: how to calibrate propensity scores for causal effect estimation.
-	The paper theoretically defines and addresses the above problem, providing convincing evidence.
-	A simple calibration method is proposed in the paper, and the necessity of calibration is demonstrated through experiments.

**Q2-3 Extent To Which Claims Are Supported By Evidence:**

3: Good: the main claims are supported by convincing evidence (in the form of adequate experimental evaluation, proofs, (pseudo-)code, references, assumptions).

**Q2-4 Reproducibility:**

3: Good: key resources (e.g. proofs, code, data) are available and key details (e.g. proofs, experimental setup) are sufficiently well-described for competent researchers to confidently reproduce the main results.

**Q3 Main Strengths:**

-	This paper investigates a critical issue: how to calibrate propensity scores for causal effect estimation.
-	The paper theoretically defines and addresses the above problem, providing convincing evidence.
-	A simple calibration method is proposed in the paper, and the necessity of calibration is demonstrated through experiments.

**Q4 Main Weakness:**

-	The lack of comparison with existing methods for causal effect calibration, such as [1] [2], which address the calibration problem of heterogeneous treatment effects that are more challenging than ATE.
-	The main contribution of this paper lies in the theoretical aspects, while the methodology (isotonic calibration) described in section 4 seems to lack sufficient technical contributions, especially compared to [1]. I would appreciate it if the authors could further clarify the technical innovations of this paper.

[1] Van Der Laan L, Ulloa-Pérez E, Carone M, et al. Causal isotonic calibration for heterogeneous treatment effects[C]//International Conference on Machine Learning. PMLR, 2023: 34831-34854.

[2] Xu Y, Yadlowsky S. Calibration error for heterogeneous treatment effects[C]//International Conference on Artificial Intelligence and Statistics. PMLR, 2022: 9280-9303.

**Q5 Detailed Comments To The Authors:**

The theoretical framework provided in this paper for propensity score calibration is a significant contribution to the causal inference community. However, I would like to understand the strengths and weaknesses of this paper compared to [1] and [2].

[1] Van Der Laan L, Ulloa-Pérez E, Carone M, et al. Causal isotonic calibration for heterogeneous treatment effects[C]//International Conference on Machine Learning. PMLR, 2023: 34831-34854.

[2] Xu Y, Yadlowsky S. Calibration error for heterogeneous treatment effects[C]//International Conference on Artificial Intelligence and Statistics. PMLR, 2022: 9280-9303.

**Q9 Complying With Reviewing Instructions:**

Yes

---

> ### Author Rebuttal · Authors · 2024-04-08
>
> We thank the reviewer for their positive comments and constructive feedback. Below we respond to the concerns and questions raised.
>
> __Comparing Against ‘Causal Isotonic Calibration’ (Van Der Laan et. al[1])__
>
> Van Der Laan et. al[1] propose causal isotonic calibration to improve the estimation of heterogeneous treatment effects (HTEs). Their work enforces a __different notion of calibration__ on the HTE prediction: The average HTE of units with a given predicted HTE is equal to the shared predicted value. The goal of their work is to ensure more directly that the predicted HTE outcome is reliable for different sub-groups of the population. Our work, on the other hand, calibrates the uncertainty outcome of the propensity score model that weighs the treated and control outcomes to achieve covariate balance. Our definition of calibration ensures that the number of units receiving treatment, given X % predicted probability of receiving treatment, is equal to X %. This definition ensures that we avoid extreme propensity weights while balancing covariates and improve the error bounds on causal effect estimates. Both calibration methods can be implemented using isotonic regression (with/without cross-validation splits to train the recalibrator).
>
> Although we only present results with the ATE metric in our main paper, __our method is applicable to both ATE and HTE estimation. Table 6 in our Appendix demonstrates that propensity score calibration also improves HTE estimation consistently in the drug effectiveness experiment from Table 1__ (by tracking the Precision in Estimation of Heterogeneous Treatment Effect).
>
> While causal isotonic regression[1] performs calibration of HTEs more directly, it requires the propensity score model or outcome model to be sufficiently accurate for their approach to work. Additionally, their experimental setup ensures that their propensity score models are consistent and do not produce extreme values. Ensuring this is difficult in practice, especially when we do not know the true treatment assignment mechanism in observational studies. Our uncertainty calibration method can be applied to mis-specified propensity models (possibly producing extreme weights) as demonstrated in several experiments. Table 7 and Table 9 in our Appendix demonstrate consistent improvement in causal effect estimates over a range of base propensity score models. Applying our method to calibrate propensity scores in HTE estimation could be an interesting way to reduce the issue with extreme propensity weights while performing causal isotonic regression[1] (e.g., in the case of high-dimensional/complex covariates).
>
> We will cite this work in the paper and discuss the comparison of our method with it.
>
> __Comparing Against ‘Calibration error for heterogeneous treatment effects’ (Xu et al[2])__
>
> Xu et al. [2] propose a technique to compute the calibration error while estimating heterogeneous treatment effects (HTEs). Again, their notion of calibration on HTE prediction means the following:  The average HTE of units with a given predicted HTE is equal to the shared predicted value. _They do not propose techniques to enforce their notion of calibration, but demonstrate the effectiveness of their method to compute calibration error as a metric while estimating HTEs_. Our work proposes methods to _enforce_ propensity score calibration and demonstrates consistent improvement in the accuracy of ATE estimates. Table 6 in our Appendix demonstrates that propensity score calibration also improves HTE estimation consistently in the drug effectiveness experiment
>
> Additionally, Xu et al. [2] require that the estimated propensity scores do not produce extreme weights for their method to work. However, it is hard to ensure this in practice, especially when the true treatment assignment mechanism is unknown in observational studies. Our method can be applied to mis-specified propensity score models that produce extreme weights. Hence, it is possible to apply our method independently to obtain calibrated propensity score before estimating the calibration error as proposed by Xu et al.
>
>
> [1] Van Der Laan L, Ulloa-Pérez E, Carone M, et al. Causal isotonic calibration for heterogeneous treatment effects[C]//International Conference on Machine Learning. PMLR, 2023: 34831-34854.
>
> [2] Xu Y, Yadlowsky S. Calibration error for heterogeneous treatment effects[C]//International Conference on Artificial Intelligence and Statistics. PMLR, 2022: 9280-9303.

---

### Official Review · Reviewer_v3iY · 2024-03-26

**Q2-1 Originality-Novelty:** 3
**Q2-2 Correctness-Technical Quality:** 3
**Q2-5 Clarity Of Writing:** 3

**Q1 Summary And Contributions:**

The paper presents a methodological advancement in causal effect estimation through the recalibration of propensity scores. Propensity scores, widely used to balance observed covariates in observational studies, are argued to require calibration—meaning, for instance, that a predictive treatment probability of 90% should indeed reflect a 90% chance of individuals being assigned to the treatment group. The authors propose simple recalibration techniques to ensure this property, demonstrate through formal arguments and empirical evaluation that calibration is necessary for unbiased treatment effect estimation (for instance when using AIPW and where the outcome regression is misspecified), and show that calibrated propensities can improve causal effect estimation across various tasks.

**Q2-3 Extent To Which Claims Are Supported By Evidence:**

3: Good: the main claims are supported by convincing evidence (in the form of adequate experimental evaluation, proofs, (pseudo-)code, references, assumptions).

**Q2-4 Reproducibility:**

3: Good: key resources (e.g. proofs, code, data) are available and key details (e.g. proofs, experimental setup) are sufficiently well-described for competent researchers to confidently reproduce the main results.

**Q3 Main Strengths:**

- The focus on recalibration of propensity scores for causal effect estimation addresses a critical but often overlooked aspect of propensity score methodology.

- The paper provides a theoretical argument for the necessity of calibration in propensity scoring models and supports these claims with empirical evidence across applications.

- By demonstrating that calibration can improve causal effect estimates and potentially reduce the need for complex model specifications, the paper offers insights for practitioners in epidemiology, economics, and other fields reliant on observational data.

**Q4 Main Weakness:**

- The empirical evaluations, while thorough, could be expanded to include a wider variety of datasets and scenarios, particularly where the assumptions about treatment assignment and outcome generation mechanisms are violated.

- The paper could benefit from a more detailed discussion on the limitations of the proposed methods, including situations where recalibration might not be effective or could introduce its own biases.

**Q5 Detailed Comments To The Authors:**

The authors have provided a compelling argument for the importance of calibration in propensity score models and its impact on causal effect estimation. The theoretical grounding and empirical evidence support the claim that calibration can mitigate issues such as overconfidence in model predictions and the high variance of estimators. Overall, I think the paper makes some compelling arguments; although I didn't carefully look at the proofs.

- The effectiveness of the recalibration techniques is demonstrated across several tasks, including high-dimensional image covariates and genome-wide association studies. How generalizable are these recalibration techniques to other domains or more complex treatment effect scenarios, such as those involving multiple treatments or longitudinal data?

- While the paper does compare calibrated propensities to uncalibrated models and mentions other methods like trimming and covariate balancing, it would be beneficial to have a more detailed comparison. How do calibrated propensity scores perform relative to methods in causal inference that directly adjust for confounding without relying on propensity scores?

- The recalibration techniques' implementation details could be expanded. For instance, how sensitive are the proposed algorithms to the choice of the recalibration model and the loss function used? Are there best practices for selecting these components based on the characteristics of the dataset or the underlying propensity score model?

**Q9 Complying With Reviewing Instructions:**

Yes

---

> ### Author Rebuttal · Authors · 2024-04-08
>
> We thank the reviewer for their positive comments and constructive feedback. Below we respond to the concerns and questions raised.
>
> __Experimenting with more complex covariates/treatments/outcomes__
>
> When working with multiple treatments, extreme propensity weights are more likely as the number of treatments increases, causing erratic IPW estimates with large sample variances[1]. Trimming is suggested as a standard technique to avoid this problem, but this also introduces bias[1]. We believe that propensity score calibration can avoid extreme propensities without introducing additional bias as long as the true propensity scores are not approaching zero. To implement calibrated propensities for multiple treatments, we need simple modifications to the recalibrator similar to the approach discussed here [2].
>
> In the case of longitudinal data, time-dependent confounding from covariates and a long historical context can make it harder to apply g-computation (direct adjustment for confounding without propensity scores) due to high dimensionality. When using inverse propensity-based estimators, extreme propensity scores are more likely due to the long context. Stabilized weights are generally used to avoid this issue. Although we work with a very simple setup in Table 1, we see that calibrated propensity scores with/without combining stabilized weights improved the ATE estimates further. As long as true propensities do not approach zero, calibrating propensity score models on longitudinal data and combining them with stabilized weights could be explored as a future direction.
>
> In Table 7 (Appendix G), we compare a range of base propensity score models where the true treatment assignment function is non-linear logical XOR (Appendix D). _We see the benefits of calibration across varying degrees of misspecification in the base model._ Table 9  (Appendix G) also demonstrates consistent improvement in causal effect estimates for high-dimensional GWAS experiments over a range of base propensity score models.
>
> We will extend Table 1 to include an experiment with multiple treatments as a simple extension of our method. We will also add the suggested regimes in future work. We thank the reviewer for these suggestions.
>
>
> [1] Kilpatrick RD, Gilbertson D, Brookhart MA, Polley E, Rothman KJ, Bradbury BD. Exploring large weight deletion and the ability to balance confounders when using inverse probability of treatment weighting in the presence of rare treatment decisions.
>
> [2] Zadrozny, B. and Elkan, C. Transforming classifier scores into accurate multiclass probability estimates.
>
>
> __Details on recalibration method. Sensitivity of method to choices like recalibration model, loss function, rules of thumb to select them__
>
> When the treatments are binary, we can choose between isotonic regression and logistic regression as the recalibrator. Since isotonic regression is prone to overfitting, we prefer to use logistic regression when the calibration dataset size is small (e.g., <1000 data points). The choice of proper loss function is dependent on the choice of recalibrator. For example, log loss is an appropriate proper loss function for learning the logistic regression recalibrator model. Leave-one-out cross-validation splits could be useful to generate the calibration dataset when the dataset size is small.
>
> When moving to the multiple treatment/ continuous treatment setup, designing the recalibrator may involve more choices (for example, we can have a simple neural network as a recalibrator in the case of continuous treatments). Using a separate cross-validation dataset would help select these hyperparameters.
>
> We will add a section in the Appendix of the paper on details of the recalibration method.
>
>
> __Limitations of Calibrated Propensities__
>
> When we apply calibrated propensities for causal effect estimation, we assume that there is no hidden confounding. In the presence of unobserved confounders, calibrating the propensity scores will not be helpful. Calibration can ensure accurate causal effect estimates when the propensity score model Q can discriminate between different treatments. For example, if the propensity model $Q(T|X) = P(T)$, then $Q$ is perfectly calibrated but cannot estimate accurate treatment effects. Ensuring that Q can discriminate between different treatments is a strong condition and we discuss this further in Appendix H.3.4.
>
> We will add a section on limitations to the final version of the paper.

---

### Meta-Review · Area_Chair_vcZP · 2024-04-11

All reviewers were favorable on the work. Reviewer 7D7j's summary captures the opinions of the reviewers well:

"The paper studies the importance of having a well-calibrated propensity score model in the context of propensity-weighted causal effect estimation. Theoretically, in the considered context, the paper: (i) shows that calibration is a necessary condition for a correct propensity score model, (ii) derives error bounds of the causal effect estimate based on the quality of the uncertainties of the propensity score model and shows that calibration improves the bound, (iii) shows that calibration can reduce variance of estimators caused by extreme propensity weights. Then, the paper proposes an algorithm for learning calibrated propensity score models, with theoretical guarantees on calibration and regret. Finally, the utility of the algorithm is demonstrated in extensive numerical studies."

A couple of reviewers pointed out the need to cite other works, and the authors have committed to citing them for the camera-ready version.

[1] Van Der Laan L, Ulloa-Pérez E, Carone M, et al. Causal isotonic calibration for heterogeneous treatment effects[C]//International Conference on Machine Learning. PMLR, 2023: 34831-34854.

[2] Xu Y, Yadlowsky S. Calibration error for heterogeneous treatment effects[C]//International Conference on Artificial Intelligence and Statistics. PMLR, 2022: 9280-9303.